# On the Representational Capacity of Recurrent Neural Language Models

**Franz Nowak**[1]  **Anej Svete**[1]  **Li Du**[2]  **Ryan Cotterell**[1]
[1]ETH Zürich    [2] Johns Hopkins University
{fnowak, asvete, rcotterell}@ethz.ch   leodu@cs.jhu.edu

## Abstract

This work investigates the computational expressivity of language models (LMs) based on recurrent neural networks (RNNs). Siegelmann and Sontag (1992) famously showed that RNNs with rational weights and hidden states and unbounded computation time are Turing complete. However, LMs define weightings over strings in addition to just (unweighted) language membership and the analysis of the computational power of RNN LMs (RLMs) should reflect this. We extend the Turing completeness result to the probabilistic case, showing how a rationally weighted RLM with unbounded computation time can simulate any deterministic *probabilistic* Turing machine (PTM) with rationally weighted transitions. Since, in practice, RLMs work in real-time, processing a symbol at every time step, we treat the above result as an upper bound on the expressivity of RLMs. We also provide a lower bound by showing that under the restriction to real-time computation, such models can simulate deterministic real-time rational PTMs.

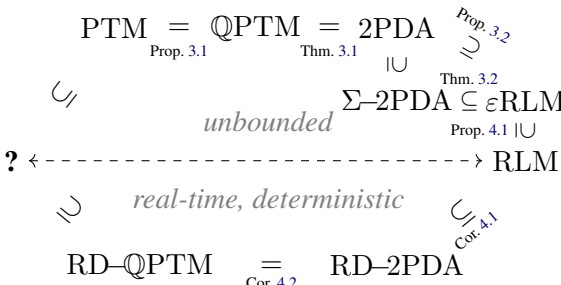
https://github.com/rycolab/
rnn-turing-completeness

## 1 Introduction

A **language model** (LM) is definitionally a semimeasure[1] over strings (Icard, 2020). Recent advances in their capabilities, leading to the widespread adoption of LMs, have sparked interest in their theoretical properties and guarantees. Previous work has characterized modern architectures such as recurrent neural networks (RNNs; Elman, 1990; Hochreiter and Schmidhuber, 1997) in terms of the formal languages they can and cannot recognize (Kleene, 1956; Minsky, 1967; Siegelmann and Sontag, 1992; Merrill et al., 2020, *inter alia*). However, characterizing LMs as formal languages is, in some sense, a category error because LMs encode semimeasures over strings instead of deciding language membership (Chen et al., 2018). In this work, we thus offer another perspective on understanding RNN LMs (RLMs) by asking: What classes

---

[1]Roughly, a semimeasure is a generalization of a probability measure over strings such that the total measure may be less than one.

$$\begin{array}{ccc} \text{PTM} & = & \mathbb{Q}\text{PTM} & = & \text{2PDA} \\ & \text{\tiny Prop. 3.1} & & \text{\tiny Thm. 3.1} & & \end{array}$$

Figure 1: Roadmap through the paper showing relations between different models of computation. A PTM is a reformulation of the classic probabilistic Turing machine. A $\mathbb{Q}$PTM is a PTM with multiple rationally weighted transition functions. A 2PDA is a probabilistic two-tape pushdown automaton. A $\Sigma$–2PDA is a 2PDA that is deterministic in its output alphabet. An RLM is a simple RNN LM. An $\varepsilon$RLM is an RLM augmented with an empty output symbol ($\varepsilon$). The prefix "RD-" denotes deterministic real-time machines.

of semimeasures over strings can RLMs represent, i.e., what is their computational expressivity?

The empirical capabilities of trained language models have spurred a large field of work testing their reasoning and linguistic abilities. However, our theoretical understanding of what these models are inherently capable of is still lacking (Deletang et al., 2023). Connecting an LM architecture to well-understood models of computation can help us determine whether the architecture is able to perform the sequences of computations required to carry out an algorithmic task (Pérez et al., 2019). Furthermore, connecting it to linguistic models can tell us whether the architecture is capable of correctly modeling the linguistic structure of a sentence symbolically (Linzen et al., 2016). Finally, characterizing the types of semimeasures the architecture can represent allows us to make more concrete claims about the abilities and limitations of the architecture itself.

RLMs have set many important milestones in language modeling and still hold the state of the art in some important settings of natural language processing (Qiu et al., 2020; Orvieto et al., 2023). Moreover, despite the recent trend towards the recurrence-free and, thus, parallelizable,

transformer-based LMs (Vaswani et al., 2017), elements of recurrence have found their way into recent language models and RNNs themselves have recently even been proposed as alternatives or extensions to some high-performing models (Peng et al., 2023; Orvieto et al., 2023; Zhou et al., 2023). At a high level, RNNs work by maintaining a hidden state encoding the processed string, much like how formal models of computation such as Turing machines process and store information. This sequential nature has motivated the comparison of the computational power of RNNs to that of various formal models of computation, from simple models such as finite-state automata (Kleene, 1956; Merrill et al., 2020) and counter machines, all the way up to Turing machines and related models (Minsky, 1967; Siegelmann and Sontag, 1992; Weiss et al., 2018).

Precisely where RNNs end up on the hierarchy of formal models of computation depends on the specific formalization. In this work, we characterize the computational power of RNNs in their most permissive formalization, i.e., one that allows RNNs to process and produce rational-valued vectors and perform an unbounded number of computational steps per input symbol by allowing them to emit empty tokens, $\varepsilon$, in between words. Siegelmann and Sontag (1992) show that such RNNs can simulate any deterministic Turing machine and are, hence, Turing complete.[2] While this sheds light on the processing power of RNNs, their result is not directly applicable to language modeling, as it does not take into account the probability assigned to the strings. By extending Siegelmann and Sontag's (1992) construction to the probabilistic case, we provide first steps towards understanding the expressive power of RLMs with rational arithmetic. We show that RLMs with rational weights and unbounded computation time can compute exactly the same semimeasures over strings as probabilistic Turing machines.

On one hand, rational arithmetic offers a reasonably faithful formalization of real-world models in that computer scientists often analyze numerical algorithms using such an idealization.[3] However,

on the other hand, the assumption of unbounded computation time *does* represent a large departure from realistic models. In practice, RLMs perform a constant number of computational steps per symbol, operating in a real-time setting (Weiss et al., 2018). Therefore, we treat the above result as an *upper bound* on the computational power of recurrent RLMs. As a lower bound, we study a second type of RLMs, restricting the models to operate in real-time, which results in a more fine-grained hierarchy of specific Turing machine-like models equivalent to an RLM. We hence characterize the expressivity of RLMs in terms of classical computational models.

Our work offers a first step towards a comprehensive characterization of the expressivity of RLMs in terms of the classes of probability measures they can represent. In addition to providing insights into the computational capacity of RLMs, the work also follows the recent exploration of the measure-theoretic foundations of LMs (Welleck et al., 2020; Meister et al., 2023; Du et al., 2023), while focusing on a particular architecture. We conclude the paper by posing several open questions on the exact position of RLMs in the hierarchy of relevant computational models. Fig. 1 shows a roadmap of the paper, with the two types of RLMs of interest and their relation to different formal computational models.

## 2 Preliminiaries

In this section, we build up the necessary definitions and vocabulary for the rest of the paper.

### 2.1 Recurrent Neural Language Models

A **formal language** $L$ is a subset of the Kleene closure $\Sigma^*$ of some finite non-empty set of symbols, i.e., an **alphabet**, $\Sigma$. An element of $\Sigma^*$ is called a **string**, $\boldsymbol{y}$. Furthermore, $\varepsilon$ denotes the empty string. We assume throughout that $\varepsilon \notin \Sigma$ and denote $\Sigma_\varepsilon \stackrel{\text{def}}{=} \Sigma \cup \{\varepsilon\}$. A (discrete) **semimeasure** over $\Sigma^*$ is a function $\mu \colon \Sigma^* \to [0, 1]$ such that $\sum_{\boldsymbol{y} \in \Sigma^*} \mu(\boldsymbol{y}) \leq 1$ (Bauwens, 2013; Icard, 2020). If the semimeasure of all strings sums to one, i.e., $\sum_{\boldsymbol{y} \in \Sigma^*} \mu(\boldsymbol{y}) = 1$, then $\mu$ is called a **probability measure**.[4] A **language model** (LM) $p$ is defined as a semimeasure over $\Sigma^*$. If $p$ is a probability measure, we call it a **tight** language model.

---

[2]Note that the restriction on the Turing machine of being deterministic does not change the generated language since any non-deterministic Turing machine has an equivalent deterministic Turing machine, albeit with a potentially much longer running time for a given string (Gill, 1974).

[3]In many cases even real arithmetic is assumed, e.g., when theoretically analyzing optimization algorithms (Forst and Hoffmann, 2010, Ch. 1).

[4]Note that our definition differs from Li and Vitányi's (2008) who instead define semimeasures over *prefix strings*.

Most modern LMs are autoregressive, meaning they define $p(\boldsymbol{y})$ through conditional semimeasures of the next symbol given the string produced so far and the measure of ending the string, i.e.,

$$p(\boldsymbol{y}) \overset{\text{def}}{=} p(\text{EOS} \mid \boldsymbol{y}) \prod_{n=1}^{N} p(y_n \mid \boldsymbol{y}_{<n}) \quad (1)$$

where EOS denotes the special end-of-string symbol, which specifies that the generation of a string has halted. The inclusion of EOS allows (but does not guarantee) a $p$ defined autoregressively to define a probability measure over $\Sigma^*$ (Du et al., 2023). We will denote $\overline{\Sigma} \overset{\text{def}}{=} \Sigma \cup \{\text{EOS}\}$.

We will use the following definition of an RNN.

**Definition 2.1.** *A **simple RNN** $\mathcal{R}$ is an RNN with the following hidden state update rule:*

$$\mathbf{h}_t \overset{\text{def}}{=} f(\mathbf{U}\mathbf{h}_{t-1} + \mathbf{V}\mathbf{r}(y_t) + \mathbf{b}) \quad (2)$$

*where $\mathbf{h}_0$ is a vector in $\mathbb{Q}^D$, $D$ is the dimensionality of the hidden state, $\mathbf{r} \colon \Sigma \to \mathbb{Q}^R$ is the symbol representation function, $R$ is the embedding dimension, $\mathbf{b} \in \mathbb{Q}^D$, $\mathbf{U} \in \mathbb{Q}^{D \times D}$, and $\mathbf{V} \in \mathbb{Q}^{D \times R}$. The function $f$ is the **saturated sigmoid**, defined as:*

$$f(x) \overset{\text{def}}{=} \begin{cases} 0 & \textbf{if } x < 0 \\ x & \textbf{if } x \in [0, 1] \\ 1 & \textbf{if } x > 1 \end{cases} \quad (3)$$

Due to their sequential nature, RNNs have been linked to formal models of computation such as finite-state automata, pushdown automata (PDA), and Turing machines under various formalizations with different implications on computational power (e.g., Siegelmann and Sontag, 1992; Hao et al., 2018; Korsky and Berwick, 2019; Merrill, 2019; Merrill et al., 2020; Hewitt et al., 2020, *inter alia*). For example, if, instead of using the saturated sigmoid, we assumed that $f$ is a function that maps to a finite set, this would result in RNNs that are at most as expressive as finite-state automata (Minsky, 1967; Svete and Cotterell, 2023). Merrill et al. (2020) study the computational power of saturated RNNs by investigating the effect of asymptotically large weights. Finally, Siegelmann and Sontag (1992) assumes rational-valued arithmetic, which is the convention we follow in this work.

An RNN specifies an LM by defining a conditional probability measure over $y_t$ given $\boldsymbol{y}_{<t}$. Let $\mathbf{E} \in \mathbb{Q}^{|\Sigma| \times D}$ be an output matrix and $\mathcal{R}$ an RNN. An RLM is an LM whose conditional probability

measures are defined by projecting $\mathbf{E}\mathbf{h}_t$ to the probability simplex $\boldsymbol{\Delta}^{|\overline{\Sigma}|-1}$ using a projection function $\boldsymbol{\pi} \colon \mathbb{Q}^D \to \boldsymbol{\Delta}^{|\overline{\Sigma}|-1}$:

$$p(y_t \mid \boldsymbol{y}_{<t}) \overset{\text{def}}{=} \boldsymbol{\pi}(\mathbf{E}\mathbf{h}_t)_{y_t} \quad (4)$$

When generating from an RLM, we assume the next symbol is sampled according to the probabilities defined by $\boldsymbol{\pi}(\mathbf{E}\mathbf{h}_t)$ and is then passed as the next input symbol back into the RNN until EOS is generated.

## 2.2 Turing Machines

We use a reformulation of the classic definition of a probabilistic Turing machine similar to Weihrauch's (2000) Type-2 Turing machine.[5]

**Definition 2.2.** *A **probabilistic Turing machine** (PTM) is a two-tape machine specified by the 6-tuple $\mathcal{M} = (Q, \Sigma, \Gamma, \delta_1, \delta_2, q_\iota, q_\varphi)$, where*

- *$Q$ is a finite set of states;*
- *$\Sigma$ and $\Gamma$ are the input and tape alphabets, and $\Gamma$ includes the blank symbol $\sqcup$;*
- *$q_\iota, q_\varphi \in Q$ are initial and final states;*
- *$\delta_{\{1,2\}} \colon Q \times \Gamma \to Q \times \Gamma \times \Sigma_\varepsilon \times \{L, R, N\}$ are two transition functions, one of which is chosen at random at each computation step.*

The Turing machine defined above has two tapes. The first is a working tape on which symbols from the tape alphabet $\Gamma$ can be read and written. The second is an append-only output tape on which $\mathcal{M}$ writes symbols of the output alphabet $\Sigma$. In the beginning, both tapes are empty, i.e., the working tape has only blank symbols $\sqcup$, and the output tape has only empty symbols $\varepsilon$. Starting in the initial state $q_\iota$, at any time step $t$, the machine samples one of the two transition functions at random, each with probability $\frac{1}{2}$, and applies it. A given transition can be written as $(q, \gamma) \xrightarrow{y/d} (q', \gamma')$, where $q, q' \in Q, \gamma, \gamma' \in \Gamma, y \in \Sigma_\varepsilon$, and $d \in \{L, R, N\}$. The semantics of such a transition is as follows: When in state $q$ and reading $\gamma$ on the working tape, go to state $q'$, write $\gamma'$ to the working tape, write $y \in \Sigma_\varepsilon$ to the output tape, and move the head on the working tape by one symbol along the tape in the direction $d$, that is, left ($L$), right ($R$), or stay in place ($N$).[6] When $y = \varepsilon$, the machine simply does

---

[5]Note that our adding an additional tape does not increase the power of the Turing machine (Sipser, 2013, Ch. 3).

[6]The stay-in-place operation is often added to make proofs simpler but does not add expressivity (Sipser, 2013, Ch. 3).

not write anything on the output tape. The machine **halts** once it reaches the final state $q_\varphi$. We call the sequence of symbols $\boldsymbol{y} \in \Sigma^*$ on the output tape at that point the **output** of the machine.

Note that, once a transition function has been chosen, since it is a function, the next transition is uniquely determined by the current state $q$ and the current tape symbol $\gamma$ under the read-write head. In the following, we call a pair of $(q, \gamma) \in Q \times \Gamma$ a **configuration** of $\mathcal{M}$.

**Remark 2.1.** *Given a probabilistic Turing machine $\mathcal{M}$ as defined above, we can get the probability of $\mathcal{M}$ halting and outputting a specific string $\boldsymbol{y}$ by summing the probabilities of all halting paths[7] through the machine that result in $\boldsymbol{y}$ being written on the output tape (the probability of each path is $2^{-n}$, where $n$ is the number of computation steps).*

Remark 2.1 induces a semimeasure over the possible sequences $\boldsymbol{y} \in \Sigma^*$ that a PTM $\mathcal{M}$ can output, which we will call $\mathbb{P}_\mathcal{M}$. That is, $\mathbb{P}_\mathcal{M}(\boldsymbol{y})$ is the probability that $\mathcal{M}$ will halt with $\boldsymbol{y}$ as its output.

**Remark 2.2.** *The notion of halting probability as defined in Remark 2.1 has a counterpart in* RLM*s, namely, the probability mass placed on all finite strings generated (Icard, 2020). For details, see Appendix A.*

## 2.3 Pushdown Automata

We now move to another probabilistic computational model: The two-stack pushdown automaton.

**Definition 2.3.** *A **probabilistic two-stack pushdown automaton** (2PDA) is a two-stack-machine defined by the tuple $\mathcal{P} = (Q, \Sigma, \Gamma, \delta, q_\iota, q_\varphi)$, where*

- *$Q$ is a finite set of states;*
- *$\Sigma$ and $\Gamma$ are the input and stack alphabets, and $\Gamma$ includes the bottom-of-stack symbol $\perp$;*
- *$q_\iota, q_\varphi \in Q$ are the initial and final states;*
- *$\delta \colon Q \times \Gamma \times \Sigma_\varepsilon \times Q \times \Gamma_\varepsilon^4 \to \mathbb{Q}$ is a transition weighting function.*

To make the connection to Turing machines more straightforward, our definition of a 2PDA assumes that its transitions depend on its current state $q$ and the top stack symbol of *only the first* of the two stacks.[8] We write transitions as $q \xrightarrow[\gamma_2 \to \gamma_4]{y, \gamma, \gamma_1 \to \gamma_3} q'$, for $q, q' \in Q$, $y \in \Sigma_\varepsilon$, $\gamma \in$

$\Gamma, \gamma_1, \gamma_2, \gamma_3, \gamma_4, \in \Gamma_\varepsilon$. Such a transition denotes that the 2PDA in the state $q$ with the symbol $\gamma$ on top of the first stack pops $\gamma_1$ and $\gamma_2$ from the first and second stack, pushes $\gamma_3$ and $\gamma_4$ onto the stacks and moves to state $q'$. At the same time, the 2PDA consumes or emits (depending on the use-case) a symbol from $\Sigma_\varepsilon$.

Again we assume the rational weighting function $\delta$ of a 2PDA is locally normalized over configurations $(q, \gamma) \in Q \times \Gamma$, where $\gamma$ is the symbol currently on the top of the first stack:

$$\sum_{\substack{y \in \Sigma_\varepsilon, q' \in Q, \\ \gamma_1, \gamma_2, \gamma_3, \gamma_4 \in \Gamma_\varepsilon}} \delta\left(q \xrightarrow[\gamma_2 \to \gamma_4]{y, \gamma, \gamma_1 \to \gamma_3} q'\right) = 1 \quad (5)$$

A 2PDA starts at the initial state $q_\iota$ with both stacks empty (only containing the symbol $\perp$) and then sequentially applies transitions according to their probability given by $\delta$. The automaton halts when reaching the final state, $q_\varphi$. The sequence of the symbols output by the automaton concatenated in the order of transitions taken constitutes the output string.

**A note on variants of computational models.** Our definition of a probabilistic Turing machine differs from the traditional definition of mere language *acceptors* in that they start from a starting state $q_\iota$ and then iteratively apply probabilistic transitions to *generate* outputs $\boldsymbol{y} \in \Sigma^*$, where each specific $\boldsymbol{y}$ has a corresponding probability of being produced. This is to simplify the comparison to 2PDA and to be able to interpret them as language models in their own right.

Next, we define what it means for two probabilistic models of computation to be equivalent.

**Definition 2.4.** *We say that two probabilistic computational models $\mathcal{M}_1$ and $\mathcal{M}_2$ are **weakly equivalent** if, for any string $\boldsymbol{y} \in \Sigma^*$, we have $\mathbb{P}_{\mathcal{M}1}(\boldsymbol{y}) = \mathbb{P}_{\mathcal{M}2}(\boldsymbol{y})$. If, furthermore, there exists a weight-preserving, yield-preserving[9] bijection between halting paths in the two models, they are called **strongly equivalent**.[10]*

## 3 An Upper Bound

In this section, we establish an upper bound on the expressive power of RLMs by extending

---

[7]A path is a finite sequence of actions performed by a computational model starting from an initial configuration. A path is *halting* if it ends in a final state.

[8]This is without loss of generality; see Appendix B.

[9]The yield of a path is the string it produces, i.e., the sequence of symbols emitted by the actions on the path.

[10]We also lift this definition to *classes* of models $(C_1, C_2)$, which are called strongly equivalent if $\forall \mathcal{M}_1 \in C_1 \exists \mathcal{M}_2 \in C_2$ such that $\mathcal{M}_1$ and $\mathcal{M}_2$ are strongly equivalent, and vice versa.

Siegelmann and Sontag's (1992) result to the probabilistic case of language models. Because we want to upper bound the power of RLMs used in practice, we will start with a more unrealistic recurrent LM which can output empty symbols ($\varepsilon$), which we denote as $\varepsilon$RLM. We first introduce a variant of probabilistic Turing machines that can have an arbitrary (finite) number of rationally valued transition functions and show that they are strongly equivalent to 2PDA (Section 3.1). We then review Siegelmann and Sontag's (1992) construction for the unweighted case (Section 3.2). Finally, we extend this construction to the probabilistic case by showing how to simulate a 2PDA with an $\varepsilon$RLM. We conclude with the observation that this results in the equivalence of PTMs and $\varepsilon$RLMs (Section 3.3).

## 3.1 Rationally Weighted PTMs

This paper considers the expressive power of RNNs with *rational* weights. To make the connection to PTMs easier, it is helpful to define a more general type of a PTM which, instead of sampling between two equally probable transition functions, can have any number of possible transitions at a given computation step, each of which has a rational probability of being applied.

**Definition 3.1.** *A **rational-valued probabilistic Turing machine** ($\mathbb{Q}$PTM) is a PTM whose transition weighting function is of the form:*

$$\delta \colon Q \times \Gamma \times \Sigma_\varepsilon \times Q \times \Gamma \times \{L, R, N\} \to \mathbb{Q} \quad (6)$$

*In other words, for any current configuration, it assigns a rational-valued probability in the interval $[0, 1]$ to each available transition. We require that the probabilities are normalized over configurations, that is, for all $q \in Q, \gamma \in \Gamma$:*

$$\sum_{\substack{y \in \Sigma_\varepsilon, \, q' \in Q, \\ \gamma' \in \Gamma, \, d \in \{L,R,N\}}} \delta\left( (q, \gamma) \xrightarrow{y/d} (q', \gamma') \right) = 1 \quad (7)$$

The original construction by Siegelmann and Sontag (1992) uses unweighted 2PDA which are equivalent to Turing machines (Hopcroft et al., 2001). We now want to show that we can simulate a PTM with an $\varepsilon$RLM in the same way, that is via *probabilistic* 2PDA as defined above. Therefore, we first show that PTMs and probabilistic 2PDA are also equivalent, in the following two propositions.

**Proposition 3.1.** *PTMs and $\mathbb{Q}$PTMs are weakly equivalent.*

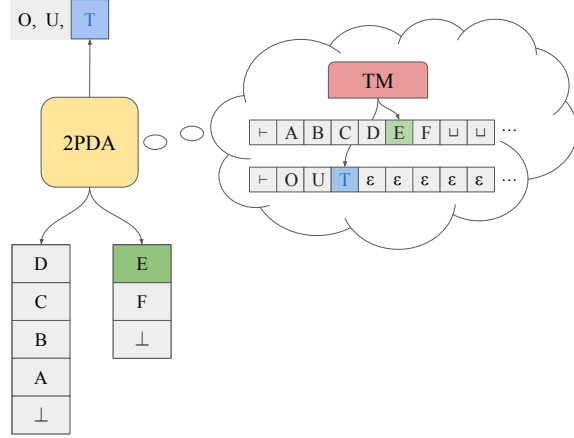

Figure 2: Simplified view of a 2PDA simulating a TM.

See Appendix C for the proof.

**Theorem 3.1.** *$\mathbb{Q}$PTMs and 2PDA are strongly equivalent.*

*Proof.* The proof that any PTM has a strongly equivalent 2PDA closely follows the proof that any (unweighted) deterministic TM can be simulated by a two-stack PDA (Thm. 8.13; Hopcroft et al., 2001). The idea is to use the two stacks in tandem to simulate the TM's infinite tape. The first stack contains the symbols to the right of the TM's head and the top symbol on the first stack is the tape symbol under the TM's head. The second stack contains the symbols to the left of the head. See Fig. 2 for a visualization. The extension to the probabilistic case, using the introduced definitions of $\mathbb{Q}$PTMs and 2PDA, is straightforward; see Appendix D. ■

## 3.2 Simulating Unweighted TMs

Before we introduce the equivalence of the models in the probabilistic case, we review the classical unweighed construction of an RNN simulating a TM first introduced by Siegelmann and Sontag (1992) and simplified by Chung and Siegelmann (2021). Specifically, Siegelmann and Sontag (1992) show that a simple RNN can encode a TM by simulating a deterministic unweighted 2PDA.[11] This 2PDA takes an input string $\boldsymbol{y}$ and maps it to the output $\mathcal{M}(\boldsymbol{y})$ given by the simulated Turing machine:

$$\mathcal{M}(\boldsymbol{y}) \stackrel{\text{def}}{=} \begin{cases} \texttt{true} & \textbf{if } \mathcal{M} \text{ accepts } \boldsymbol{y} \\ \texttt{false} & \textbf{if } \mathcal{M} \text{ rejects } \boldsymbol{y} \\ \texttt{undef} & \textbf{if } \mathcal{M} \text{ does not halt on } \boldsymbol{y} \end{cases} \quad (8)$$

---

[11]Naturally, as Thm. 3.1 suggests, *unweighted* two-stack PDAs are equivalent to unweighted TMs (Sipser, 2013).

Given a deterministic unweighted 2PDA, the construction defines an RNN that halts and stores the acceptance of $\boldsymbol{y}$ by the 2PDA in a specific neuron of the RNN, or never halts if $\mathcal{M}(\boldsymbol{y}) = \texttt{undef}$.

The crux of the construction lies in encoding the content of a stack in a neuron.[12] Importantly, the encoding must be such that *(i)* the tops of the stacks can easily be read and *(ii)* the encoding of the stack can easily be updated upon popping off or pushing onto the stack. This can, for example, be achieved by mapping a (binary) string $\boldsymbol{\gamma} = \gamma_1 \ldots \gamma_N$ into $\eta(\boldsymbol{\gamma}) \overset{\text{def}}{=} 0.\eta(\gamma_N) \ldots \eta(\gamma_1)$, where:[13]

$$\eta(\gamma) \overset{\text{def}}{=} \begin{cases} 1 & \textbf{if } \gamma = 0 \\ 3 & \textbf{otherwise} \end{cases} \quad (9)$$

Notice the opposite orientation of the two encodings: The top of the stack in $\boldsymbol{\gamma}$ is written on the right-hand side while it is the left-most digit in the numerical encoding which enables easy updates to the encoding; with this, popping $\gamma_N = 0$ can, for example, be performed by computing $f(10 \cdot \eta(\boldsymbol{\gamma}) - 1)$, popping $\gamma_N = 1$ by computing $f(10 \cdot \eta(\boldsymbol{\gamma}) - 3)$, pushing $\gamma = 0$ by computing $f\left(\frac{1}{10} \cdot \eta(\boldsymbol{\gamma}) + \frac{1}{10}\right)$, and pushing $\gamma = 1$ by computing $f\left(\frac{1}{10} \cdot \eta(\boldsymbol{\gamma}) + \frac{3}{10}\right)$.[14]

Similarly, the current state of a 2PDA is stored in a set of neurons keeping the one-hot encoding of the state, which is updated by simulating the transition function of the 2PDA. This can be done by intersecting the states reachable from the current configuration of the 2PDA and the states reachable by the currently read symbol, the same way as in the classical Minsky construction of a simple RNN simulating a finite-state automaton (Minsky, 1954). Because of the determinism of the transition function, this results in a single possible next state. The intersection can be implemented using conjunction, which is possible using the saturated sigmoid function.

---

[12] A neuron is a term of art for a component of the hidden state vector.

[13] Note that the stack encoding defined by Siegelmann and Sontag (1992) is somewhat different since it does not use the base-ten but rather base-four encoding. To keep the presentation simpler, we choose the base-ten one; the intuition remains the same.

[14] Since we are using a simple RNN, the coefficient cannot be chosen based on the current input. This is why *all* such actions are performed in parallel (into individual processing neurons). The result of the correct operation (based on the input symbol and the current stack configuration) can then be copied back into the stack neuron. This is why multiple RNN update sub-steps are required.

With this, an RNN simulating a 2PDA can be constructed by keeping a hidden state vector divided into multiple sets of values, three of which will be relevant for our extension: (1) Two **stack neurons**, each representing a stack; (2) Two **readout neurons**, each encoding the symbols on top of one of the stacks; (3) $|Q|$ **state neurons** encoding the current state of the 2PDA. The readout neurons can be computed from the stack encodings $\eta(\boldsymbol{\gamma}_1)$ and $\eta(\boldsymbol{\gamma}_2)$ similarly to how the stack encodings are updated. See top of Fig. 3 for an illustration of how these components can be used to determine the quantities relevant to determining the next action of the 2PDA. More details of the construction can be found in Chung and Siegelmann (2021, Thm. 1).

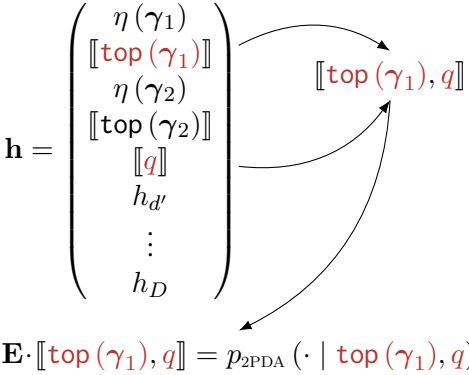

Figure 3: A schematic illustration of how the model from Chung and Siegelmann (2021) stores the information about the configuration of the 2PDA and how it can be used to access the information needed for defining string probabilities. We denote with $[\![\cdot]\!]$ the one-hot encoding function of the input arguments. $h_{d'} \ldots h_D$ refer to the rest of the hidden state not directly relevant for determining the configuration of the 2PDA.

Importantly, note that in this case, the RNN (and the 2PDA) can be fully deterministic due to the equivalence of deterministic and non-deterministic unweighted TMs. They also do not have to consider any $\varepsilon$-transitions or steps generating $\varepsilon$'s since there is no generation in the sense of Section 2.3. These two aspects will, however, require more attention in the probabilistic case, which we discuss next.

### 3.3 Simulating PTMs

A TM can perform an unbounded number of computational steps per output symbol. To account for this with RLMs in the language modeling setting, we extend their definition to one that allows generating $\varepsilon$'s, effectively allowing RNNs to perform computations without affecting the output string ($\varepsilon$RLM).

**Definition 3.2.** *An* RLM *with* $\varepsilon$*-transitions* *($\varepsilon$RLM) is an* RLM *that can output $\varepsilon$-symbols.*

More precisely, an $\varepsilon$RLM defines a symbol representation function $\mathbf{r}\colon \Sigma_\varepsilon \to \mathbb{Q}^R$ and the output matrix $\mathbf{E} \in \mathbb{Q}^{|\overline{\Sigma}_\varepsilon| \times D}$, where $D$ and $R$ are parameters depending on the 2PDA (Chung and Siegelmann, 2021), and $\overline{\Sigma}_\varepsilon = \overline{\Sigma} \cup \{\varepsilon\}$. The $\varepsilon$-symbols represent empty substrings, so the final output of the $\varepsilon$RLM is the output string with $\varepsilon$'s removed. Effectively, this gives an $\varepsilon$RLM the possibility to perform an arbitrary number of computations per symbol of the string. With this additional gadget, we are able to state our main result establishing a close connection between PTMs and $\varepsilon$RLMs.

**On determinism.** The construction we describe in the following theorem requires that the next transition of the 2PDA is fully specified given the current state $(q, \gamma)$ and the (sampled) output symbol from $\overline{\Sigma}_\varepsilon$.[15] That is, the non-determinism of the simulated 2PDA is constrained to the sampling step of the RLM, meaning there can only be one possible transition in the 2PDA per output symbol. We call a 2PDA or $\mathbb{Q}$PTM that has this property $\Sigma$**-deterministic**. Note that this is still a non-deterministic automaton; see Def. 4.1.

**Theorem 3.2** (Informal). *For every $\Sigma$-deterministic probabilistic* 2PDA*, there is a strongly equivalent $\varepsilon$RLM.*

*Rough idea.* Given a $\Sigma$-deterministic 2PDA $\mathcal{P}$, we design an $\varepsilon$RLM $\mathcal{R}$ that simulates $\mathcal{P}$ by executing its transitions and hence defining the same semimeasure over strings. We use the LM controller from Chung and Siegelmann (2021) (with the same definitions of the parameters $\mathbf{U}$, $\mathbf{V}$, and $\mathbf{b}$), as it conveniently models the transitions of $\mathcal{P}$ and exposes the parts of its configuration required to define the transition (and with it the string) probabilities. Note the additional $\varepsilon$'s do not change the construction. This leaves us with the task of appropriately defining the output matrix $\mathbf{E}$. Exposing the symbols on the top of the stacks, $\text{top}(\boldsymbol{\gamma}_1)$ and $\text{top}(\boldsymbol{\gamma}_2)$, and the current state $q$ of $\mathcal{P}$ in $\mathbf{h}_t$ (cf. Fig. 3) allow us to easily access the appropriate probabilities encoded in the output

matrix $\mathbf{E}$. Note that due to the $\Sigma$-determinism of $\mathcal{P}$, a single pair of the stack symbol and the current state $(\text{top}(\boldsymbol{\gamma}_1), q)$ determines the conditional probability measure over the next symbol.[16] More precisely, we define $\mathbf{E} \in \mathbb{Q}^{|\overline{\Sigma}_\varepsilon| \times |\Gamma_\varepsilon||Q|}$ which maps the one-hot encoding of the pair $(\gamma, q)$ to a $|\overline{\Sigma}_\varepsilon|$-dimensional vector of probabilities over the next symbol.[17] To achieve that, simply let $E_{y,(\gamma,q)}$ correspond to $\delta\left(q \xrightarrow[\circ \to \circ]{y, \gamma, \circ \to \circ} \circ\right)$, where we index the output matrix directly with the elements for cleaner notation.[18] Denoting with $[\![\gamma, q]\!]$ the one-hot encoding of the tuple $(\gamma, q)$, the vectors $\mathbf{E}[\![\gamma, q]\!]$ represent semimeasures over $\overline{\Sigma}_\varepsilon$, and $\boldsymbol{\pi}$ can be set to the identity function.[19] Considering that $\mathcal{R}$ directly simulates all possible paths of $\mathcal{P}$, it is easy to see that $\mathcal{R}$ generates a string $\boldsymbol{y}$ with a sequence of actions if and only if $\mathcal{P}$ generates it as well. Moreover, the encoding of the probabilities in $\mathbf{E}$ means that the probabilities of the action sequences are always the same. ∎

A formal statement of the theorem is given in Appendix E. We provide a proof where we show that the correspondence between the paths produced by Chung and Siegelmann's (2021) construction and the definition of $\mathbf{E}$ as described above results in a trivial weight- and yield-preserving mapping between the paths of the 2PDA $\mathcal{P}$ and the $\varepsilon$RLM $\mathcal{R}$ that simulates it. Together, this shows that the two machines are strongly equivalent.[20] This construction is implemented in `https://github.com/rycolab/rnn-turing-completeness`.

Finally, we can show that the expressivity of $\varepsilon$RLMs is bounded from above by that of a 2PDA.

---

[15] Note that this does *not* make the resulting 2PDA unambiguous in the sense that any given string can only be produced one way. There is still the potential for ambiguity because, in a given configuration, the 2PDA could either produce a certain symbol $y$, or it could move to a different configuration via an $\varepsilon$-transition and then produce the same symbol $y$ via a different transition.

[16] Recall that the target configuration of $\mathcal{P}$ depends only on the top symbol of the first stack, $\gamma \stackrel{\text{def}}{=} \text{top}(\boldsymbol{\gamma}_1)$.

[17] One-hot encodings of the state-stack symbol pairs can be obtained by applying the RNN update (sub-)step in which the nonlinearity is used to implement conjunction. This adds another sub-step to the simulation of the full 2PDA update step.

[18] Again, due to the assumed determinism, the elements with $\circ$ are irrelevant for the weights.

[19] The identity function is generally *not* a projection function onto the probability simplex. However, since its inputs in this case already lie on the probability simplex, its use is possible. More generally, we could use the sparsemax function (Martins and Astudillo, 2016), which acts like the identity function on the probability simplex. Alternatively, we could use the more popular softmax function and set the entries of $\mathbf{E}$ to the logarithms of the original probabilities (defining $\log 0 \stackrel{\text{def}}{=} -\infty$).

[20] Note that the bijection between paths is trivial in the case of a real-time $\Sigma$-deterministic 2PDA because there is no ambiguity over output symbols for any given configuration.

**Proposition 3.2.** *Every $\varepsilon$RLM has a weakly equivalent 2PDA.*

*Proof.* For the proof, see Appendix G. ∎

## 4 A Lower Bound

While Thm. 3.2 establishes a concrete result on the expressive power of $\varepsilon$RLMs, the result follows from somewhat unrealistic assumptions, namely rationally weighted networks and unbounded computation time. We contend the first assumption is a reasonable approximation, since even for small neural networks the number of expressible states can be large; assuming double precision floating point numbers, an RNN can yield as many as $2^{64 \cdot D}$ different states, where $D$ is the number of neurons.[21] However, RLMs used in practice operate in real-time, outputting a symbol at every computation step. To make our analysis closer to this use case, in this section, we develop a lower bound on the expressivity of an RLM under the real-time restriction while still allowing rational arithmetic operations.

### 4.1 Real-time RLMs

Now, we switch back to studying the more common RLM with an RNN controller based on Def. 2.1. Firstly, note that the class of RLMs is a subset of the class of $\varepsilon$RLMs:

**Proposition 4.1.** *For every RLM there exists a strongly equivalent $\varepsilon$RLM.*

*Proof.* This result follows trivially from Def. 3.2: An RLM is simply an $\varepsilon$RLM that always assigns probability 0 to outputting $\varepsilon$. ∎

### 4.2 Real-time Deterministic 2PDA

The lack of $\varepsilon$-transitions requires the properties of the simulated model to change: As in Thm. 3.2, the RNN construction requires that there is only one transition for every output symbol and configuration. Previously, this was done by imposing $\Sigma$-determinism, where non-determinism over symbols at a given time step can be reintroduced by delaying transitions through the use of additional $\varepsilon$-transitions, which is not possible here. In fact, the lack of $\varepsilon$'s and binarization means the resulting PDA has to be not just real-time, but also deterministic. We define such a 2PDA analogously to the single stack case:[22]

**Definition 4.1.** *A 2PDA is **deterministic** if:*

- *For any current state $q \in Q$, current top stack symbol $\gamma \in \Gamma$, and a given output symbol $y \in \Sigma_\varepsilon$, there is at most one transition with non-zero probability.*
- *If, in a given computation step, the weight of an $\varepsilon$-transition is non-zero, then its weight is 1 and the weight of all other transitions is 0.*

*If $\delta(q, \varepsilon, \gamma) = 0$ for all $q \in Q, \gamma \in \Gamma$ then we say it is a **real-time deterministic probabilistic** 2PDA (RD–2PDA).*

**Corollary 4.1.** *RLMs can simulate RD–2PDA.*

*Proof.* This follows directly from Thm. 3.2 since an RD–2PDA is just a special case of the 2PDA without $\varepsilon$-transitions which is exactly the restriction imposed on the RLM. ∎

### 4.3 Real-time Deterministic $\mathbb{Q}$PTM

As before, we want to connect the RLM with the better-understood PTM. To do so, we introduce a new class of rationally weighted PTMs that are deterministic and operate in real-time.

**Definition 4.2.** *A $\mathbb{Q}$PTM is **deterministic** if, for any configuration $q, \gamma \in Q \times \Gamma$, and any symbol $y \in \Sigma_\varepsilon$, there is at most one transition starting at that configuration and emitting $y$ with non-zero probability. Furthermore, if there is a transition starting in $(q, \gamma)$ outputting $\varepsilon$ with non-zero probability, it must be the only possible transition in that configuration. If there are no $\varepsilon$-transitions with non-zero probability at all, then it is called **real-time** (RD–$\mathbb{Q}$PTM).*

**Corollary 4.2.** *RD–$\mathbb{Q}$PTMs are strongly equivalent to RD–2PDA.*

*Proof.* This directly follows from Thm. 3.1 because RD–$\mathbb{Q}$PTMs are a special case of general $\mathbb{Q}$PTMs. ∎

The resulting Turing machine that our RLM can simulate is now strictly less expressive than the original PTM. See Appendix F for the proof. Hence, our lower bound is strictly less powerful than the upper bound.

---

[21]To use the state space with limited precision more effectively, one could add more stack-encoding neurons or choose more efficient encodings of the stack contents.

[22]Real-time deterministic PDA have previously been investigated (Harrison and Havel, 1972; Pittl and Yehudai, 1983, *inter alia*), but to the authors' knowledge, this has not been extended to the two-stack case.

## 5 Open Questions

This work establishes upper and lower bounds on the expressive power of RLMs. While this shows how powerful RLMs can be, the bounds do not completely and precisely characterize the models of interest. A natural question for follow-up work is, therefore, the following.

**Open Question 5.1.** *What is the exact computational power of a rationally weighted RLM?*

While we do not answer this question definitively, we hope that the steps and framework outlined here help follow-up work to establish more precise descriptions of LMs in general, be it in the form of RNNs or other architectures. Furthermore, in this work, we have introduced novel models of probabilistic computation (RD–2PDA, RD–$\mathbb{Q}$PTM) that prove useful for describing RLMs in a formal setting due to the close connection between their dynamics and those of RNNs. We also provide a preliminary analysis of the concrete computational power of the novel models. For example, in Appendix F, we provide an example of a language that can be generated by a $\mathbb{Q}$PTM but not by its real-time deterministic counterpart, thereby showing that the former is more powerful than the latter. However, we leave a more precise characterization of their expressive power to future work, specifically.

**Open Question 5.2.** *What is the relationship between deterministic $\mathbb{Q}$PTMs and non-deterministic devices lower on the hierarchy of computational models, e.g., non-deterministic probabilistic finite-state automata which cannot be represented by deterministic finite-state automata (Mohri, 1997; Buchsbaum et al., 2000)?*

**Open Question 5.3.** *Are the $\varepsilon$RLM introduced in Def. 3.2 weakly equivalent to non-deterministic $\mathbb{Q}$PTMs without the need to introduce two different types of $\varepsilon$ symbol to store the direction of the head in the outputs?*

## 6 Discussion and Conclusion

The widespread deployment of LMs in more and more far-reaching applications motivates a precise theoretical understanding of their abilities and shortcomings. In this paper, we show that tools from formal language theory, namely, probabilistic Turing machines and their extensions, offer a fruitful means of investigating those abilities by allowing us to directly characterize the classes of (probabilistic) languages LMs can represent.

Concretely, we place two different formalizations of RLMs into the framework of probabilistic Turing machines, thus characterizing their computational power. To connect our results with the bigger picture of understanding RLMs, consider again Fig. 1. The upper part of Fig. 1 (left to right) expresses the equivalence of PTMs, their rationally valued equivalent and probabilistic two-stack PDAs. These provide an upper bound for $\Sigma$-deterministic 2PDA, which are 2PDA that are deterministic in their output alphabet. These in turn can be simulated by RLMs that can output $\varepsilon$, allowing the model to perform an unbounded number of computations in between outputting output tokens. In Appendix G, we show that any $\varepsilon$RLM is weakly equivalent to some 2PDA, meaning the expressivity of $\varepsilon$RLM is upper-bounded by that of 2PDA. The lower half of the Fig. 1 shows the results on the more realistic real-time RLMs with rational weight. We show that such models match the expressive power of real-time probabilistic Turing machines with rational weights (lower left) through their correspondence to real-time deterministic probabilistic 2-stack PDAs (lower right). These results provide a set of first insights into the modeling power of modern language models and hopefully provide a starting point for the investigation of other modern architectures, such as transformers (Vaswani et al., 2017).

## Limitations

Here, we list several points of our analysis that we consider limiting. Similarly to Siegelmann and Sontag (1992), all our results assume the RLMs to have rationally valued weights and hidden states, which is not the case for RLMs implemented in practice. It remains to be shown if the bounded precision in practical implementations proves to be too restrictive for the LMs to learn to solve algorithmic problems. The upper bound result additionally assumes that computation time is unbounded, which is a departure from how RNNs function in practice. It is not clear how an RNN could be trained in a non-real-time manner, or if that would actually lead to better results on any of the standard NLP tasks.

Importantly, note that the lower bound result is likely not tight, as, we only show the ability of RLMs to simulate a specific computational model (namely, RD–$\mathbb{Q}$PTMs). There might be more

expressive models that can also be simulated by RLMs. In general, the results presented are theoretical in nature and not necessarily a practically efficient way of simulating Turing machines. Moreover, we do not suggest that trained RNNs in practice actually implement such mechanisms, but only that they are *theoretically capable* of doing it; The construction thus serves the specific purpose of theoretically simulating PTMs and does not naturally extend to training and inference outside of problems specifically designed for Turing machines.

## Ethics Statement

Our work sheds light on the theoretical capabilities of language models. To the best of the authors' knowledge, it does not pose any ethical issues.

## Acknowledgements

We would like to thank the anonymous reviewers for their helpful comments. We would also like to thank Abra Ganz and Clément Jambon for their thoughtful feedback and suggestions.

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

## A  Halting Probability and RLM

As discussed in remark 2.1, the halting probability of a PTM is defined as the sum of the probabilities of all halting paths, i.e., paths that end with $q_\varphi$. Note that EOS in a RLM and the final state $q_\varphi$ in a PTM are similar constructs, and so we can consider a similar notion for RLM.

We first see how the corresponding notion of halting probability arises in the context of RLM. While it is possible to define a probability measure over $\Sigma^*$ with the autoregressive parameterization as in Eq. (1), not all semimeasures defined by Eq. (1) are probability measures over $\Sigma^*$. For example, in the definition of RLM (Section 2.1), if we pathologically choose a projection function $\pi$ such that it always places zero probability on EOS, we would end up with a semimeasure that places 0 probability on $\Sigma^*$.[23] In fact, Eq. (1) defines a probability measure over the set of finite *and infinite* strings: $\Sigma^* \cup \Sigma^\infty$. Under this formulation, the halting probability of an RLM is the probability mass placed on the set of finite strings, $\Sigma^*$.[24] We recognize that a similar situation exists in the case of a PTM, where the non-halting trajectories are infinite sequences that can be considered as elements of $\{0, 1\}^\infty$.[25]

In trying to measure the probability mass placed on the set $\Sigma^*$ within $\Sigma^* \cup \Sigma^\infty$, we first need to define an appropriate probability measure over $\Sigma^* \cup \Sigma^\infty$. However, defining probability measures over uncountable sets such as $\Sigma^\infty$ or $\{0, 1\}^\infty$ raises nontrivial difficulties. As a simple illustration, consider an infinite fair coin toss. The sample space for this semimeasure is $\{H, T\}^\infty$. Clearly, each single infinite event (a binary string $\omega$) has probability $(\frac{1}{2})^\infty = 0$. However, treating uncountable semimeasures carelessly would result in the following paradox:

$$1 = p(\{H, T\}^\infty) = p\left( \bigcup_{\omega \in \{H,T\}^\infty} \{\omega\} \right) = \sum_{\omega \in \{H,T\}^\infty} p(\{\omega\}) = \sum_{\omega \in \{H,T\}^\infty} 0 \overset{?}{=} 0 \tag{10}$$

For reasons like this, a rigorous discussion of PTM (Roy, 2011) or RLM (Du et al., 2023) will involve a modicum of measure theory and typically starts with defining the appropriate $\sigma$-algebra. In this work, we find that introducing such technical machinery obscures our purposes and we therefore intentionally omitted them. For a rigorous discussion on the corresponding definition of halting probability in RLMs and more general autoregressive models, see Du et al. (2023).

## B  Versions of Probabilistic Two-stack Pushdown Automata

In our work, we use an adaptation of the traditional two-stack PDA whose transition function depends only on the top symbol of one of the stacks (and the current state), whereas usually the top symbols of both stacks are taken into account. This setup follows the proof by (Hopcroft et al., 2001) but warrants additional justification when applied to the probabilistic case.

**Proposition B.1.** *A* 2PDA *whose transition weighting function depends on the top symbol of both stacks can be simulated by a* 2PDA *whose transition function depends only on the top symbol of the first stack.*

*Proof.* ( $\implies$ ) Let $\mathcal{P}_1$ be a 2PDA whose transition function has the form $\delta : Q \times \Gamma^2 \times \Sigma_\varepsilon \times Q \times \Gamma^4 \to \mathbb{Q}$, such that:

$$\sum_{\substack{y \in \Sigma_\varepsilon, q' \in Q, \\ \gamma_1, \gamma_2, \gamma_3, \gamma_4 \in \Gamma_\varepsilon}} \delta\left( q \xrightarrow[\gamma_2 \to \gamma_4]{y, \gamma, \gamma', \gamma_1 \to \gamma_3} q' \right) = 1 \tag{11}$$

We now show that we can construct a 2PDA $\mathcal{P}_2$ as defined in Def. 2.3, such that for any transition in $\mathcal{P}_1$, $\mathcal{P}_2$ has a finite sequence of transitions resulting in the same state and stack configurations. Let $q \xrightarrow[\gamma_2 \to \gamma_4]{y, \gamma, \gamma', \gamma_1 \to \gamma_3} q'$ be such a transition in $\mathcal{P}_1$, where $\gamma$ is the top symbol on the first stack and $\gamma'$ is the

---

[23]For other examples where the autoregressive factorization results in $\Sigma^*$ receiving $< 1$ probability mass, see Du et al. (2023).

[24]In the case of $\Sigma^\infty$ having probability 0, we say that Eq. (1) defines a **tight** language model. In a tight language model, the halting probability is 1 and we can treat Eq. (1) as a probability measure over $\Sigma^*$.

[25]We can view each random branching as corresponding to either 0 or 1, thus resulting in an infinite binary string.

top symbol on the second stack. We can simulate this transition in $\mathcal{P}_2$ through the following chain of transitions, where we introduce a transition-specific new state $q''$:

$$q \xrightarrow[\gamma_2 \to \varepsilon]{\varepsilon, \gamma, \gamma_1 \to \gamma'} q'', q'' \xrightarrow[\varepsilon \to \gamma_4]{y, \gamma', \gamma' \to \gamma_3} q' \tag{12}$$

( $\impliedby$ ) The converse direction of proving that any such 2PDA $\mathcal{P}_1$ whose transitions depend on the top symbols of both stacks can simulate a specific 2PDA $\mathcal{P}_2$ whose transitions depend only on the top symbol of the first stack is trivial: For any transition in $\mathcal{P}_2$, we can just create a transition with the same semantics and weight 1 in $\mathcal{P}_1$ for each second stack top symbol $\gamma'$. ∎

## C Rationally Weighted Probabilistic Turing Machines

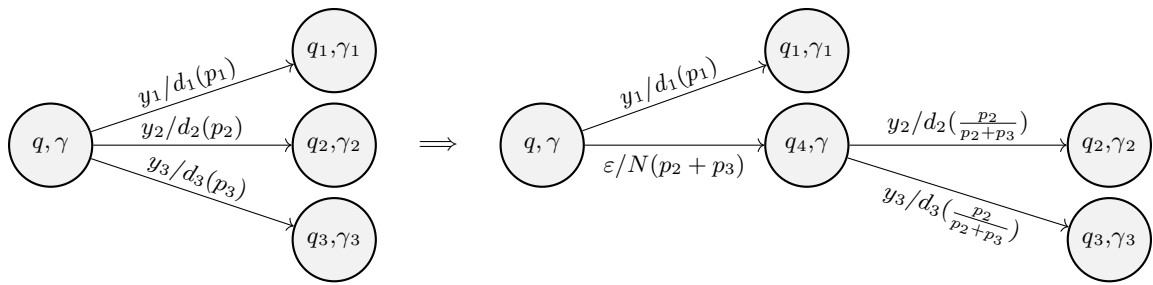

Figure 4: Binarization of an example computation graph as described in Prop. 3.1.

**Proposition 3.1.** PTM*s and* $\mathbb{Q}$PTM*s are weakly equivalent.*

*Proof.* ( $\implies$ ) The forward direction is trivial: Every PTM is a $\mathbb{Q}$PTM because $\frac{1}{2}$ is a rational number.
   ( $\impliedby$ ) We start by noting that we can transform any $\mathbb{Q}$PTM $\mathcal{M}$ into one that has exactly two possible (rational-valued) transitions at any current state and tape symbol. We do this by repeatedly applying the following transformations: For any $(q, \gamma) \in Q \times \Gamma$ that has only one possible transition, its probability is 1, so we can split it into two new identical transitions with probability $\frac{1}{2}$. For $(q, \gamma) \in Q \times \Gamma$, this allows exactly 2 possible transitions, this is already as required by the PTM (save for the probabilities, which we deal with in the next step). For any $(q, \gamma) \in Q \times \Gamma$ that allow $k > 2$ possible transitions, we repeatedly apply the following steps:

1. We choose one of the transitions whose probability we denote by $p$, and leave it as it is;
2. We then create a new $\varepsilon$-transition with $d = N$ to a new state with probability $1-p$, leaving $\gamma$ the same;
3. We then change the remaining $k - 1$ transitions to start at the new state and tape symbol.

These transformations yield a $\mathbb{Q}$PTM with a completely binarized transition function. An example of this is shown in Fig. 4. Now note that any locally normalized pair of transitions with rational weights can be replaced by a sequence of transitions whose probabilities are $\frac{1}{2}$ each (Knuth and Yao, 1976; Icard, 2020).[26] This, in conjunction with the previous transformation, allows us to convert our $\mathbb{Q}$PTM into a PTM without changing the string probability. ∎

## D Proof of Thm. 3.1

**Theorem 3.1.** $\mathbb{Q}$PTM*s and* 2PDA *are strongly equivalent.*

*Proof.* ( $\implies$ ) We want to show that for every $\mathbb{Q}$PTM, we can construct a strongly equivalent 2PDA. We do this in two steps: (1) we construct a candidate 2PDA and (2) we outline a weight- and yield-preserving bijection between accepting paths, whose existence proves strong equivalence.

---

[26]This step prevents this proof strategy from demonstrating strong equivalence in the backward direction because introduces path ambiguity. For this reason, we suspect that PTMs and $\mathbb{Q}$PTMs are in fact *not* strongly equivalent.

**Construction of $\mathcal{P}$.** Let $\mathcal{M} = (Q, \Sigma, \Gamma, \delta_{\mathcal{M}}, q_\iota, q_\varphi)$ be a $\mathbb{Q}$PTM. The constructed 2PDA $\mathcal{P} = (Q_{\mathcal{P}}, \Sigma, \Gamma, \delta_{\mathcal{P}}, q_\iota, q_\varphi)$ will inherit $\mathcal{M}$'s alphabets. It will also inherit all of $\mathcal{M}$'s states and add a number of additional ones, i.e., $Q_{\mathcal{P}} = Q \cup Q'$. As we showcase later, the subset $Q \subseteq Q_{\mathcal{P}}$ will be crucial in the analysis of the relationship between the two models. The states of $\mathcal{P}$ will be a *super*set of those in $\mathcal{M}$. For each transition in $\mathcal{M}$, we define a transition in $\mathcal{P}$ as follows, depending on the direction the head moves after writing to the tape:

(i) **Transitions that leave the head in place**, that is, transitions of the form $(q, \gamma) \xrightarrow{y/N} (q', \gamma')$. For each such transition in $\mathcal{M}$, we add an equally weighted new transition $q \xrightarrow[\varepsilon \to \varepsilon]{y, \gamma, \gamma \to \gamma'} q$ to $\mathcal{P}$.

(ii) **Transitions that move the head to the right**, i.e., $(q, \gamma) \xrightarrow{y/R} (q', \gamma')$. For each such transition in $\mathcal{M}$, we add an equally weighted new transition $q \xrightarrow[\varepsilon \to \gamma']{y, \gamma, \gamma \to \varepsilon} q'$ to $\mathcal{P}$.

(iii) **Transitions that move the head to the left**, that is, transitions of the form $\tau = (q, \gamma) \xrightarrow{y/L} (q', \gamma')$. For any such transition $\tau$ in $\mathcal{M}$, we add an equally weighted transition $q \xrightarrow[\varepsilon \to \varepsilon]{y, \gamma, \gamma \to \gamma'} q_\tau$, followed by a transition $q_\tau \xrightarrow[\gamma_2 \to \varepsilon]{\varepsilon, \gamma, \varepsilon \to \gamma_2} q$ with weight 1 to $\mathcal{P}$. Here, $q_\tau$ is a new state unique to transition $\tau$, and $\gamma_2$ is the symbol at the top of the second stack before the transition.

**A weight- and yield-preserving bijection.** Given the construction outlined above, we now show the existence of a weight- and yield-preserving bijection between the paths of the two models. While we are only interested in halting (accepting) paths, i.e., paths going from $q_\iota$ to $q_\varphi$ with a non-zero weight,[27] we prove a stronger result that there exists a weight- and yield-preserving bijection between all $Q$-subpaths, a notion we will define below.

**Definition D.1.** *A $Q$-subpath in $\mathcal{P}$ is a subpath whose last state corresponds to a state that also exists in the original $\mathcal{M}$. Recall that $\mathcal{M}$'s states are constructed to be a sub*set *of those in $\mathcal{P}$.*

($\Rightarrow$) We will define a weight- and yield-preserving mapping $\psi_1$ from the subpaths $\boldsymbol{\pi}_{\mathcal{M}}$ in the original $\mathcal{M}$ to the $Q$-subpaths $\boldsymbol{\pi}_{\mathcal{P}}$ in the $\mathcal{P}$. Fix an arbitrary subpath $\boldsymbol{\pi}_{\mathcal{M}}$ in $\mathcal{M}$. Our proof proceeds by structural induction on the subpath relation: $\boldsymbol{\sigma}_{\mathcal{M}} \prec \boldsymbol{\pi}_{\mathcal{M}} \iff \boldsymbol{\sigma}_{\mathcal{M}}$ is a strict subpath of $\boldsymbol{\pi}_{\mathcal{M}}$. Note that this is a well-founded ordering, making it suitable for induction.

**Inductive Hypothesis.** For all $\boldsymbol{\sigma}_{\mathcal{M}} \prec \boldsymbol{\pi}_{\mathcal{M}}$, the function $\psi_1$ is weight- and yield-preserving.

**Base Case.** The function $\psi_1$ is defined to map the empty subpath in $\mathcal{M}$ to the empty subpath in $\mathcal{P}$ with weight 1 and yield $\varepsilon$. This preserves the weight and yield.

**Inductive Case.** Let $\boldsymbol{\pi}_{\mathcal{M}}$ be an arbitrary non-empty subpath in $\mathcal{M}$. As depicted in Fig. 5, let $\boldsymbol{\sigma}_{\mathcal{M}}$ be the (strict) subpath of $\boldsymbol{\pi}_{\mathcal{M}}$ that omits the last transition $\tau_{\mathcal{M}}$. Then $\boldsymbol{\sigma}_{\mathcal{M}}$ is mapped as follows: $\boldsymbol{\sigma}_{\mathcal{M}} \mapsto \psi_1(\boldsymbol{\sigma}_{\mathcal{M}})$. This mapping exists and is weight- and yield-preserving by the inductive hypothesis. Now, we seek to extend the function $\psi_1$ to $\boldsymbol{\pi}_{\mathcal{M}} = \boldsymbol{\sigma}_{\mathcal{M}} \circ \tau_{\mathcal{M}}$. By our construction of $\mathcal{P}$, each of the transitions in $\boldsymbol{\pi}_{\mathcal{M}}$ is simulated either by a single transition in $\mathcal{P}$ (case (i) and case (ii)) or two consecutive transitions in $\mathcal{P}$ (case (iii)). In the two-transition case, the sequence of two actions is *unique* because the second action has a transition-dependent state name, e.g., $q_{\tau_{\mathcal{M}}}$. Thus, we can extend $\psi_1$ in the following manner:

$$\psi_1(\boldsymbol{\pi}_{\mathcal{M}}) = \psi_1(\boldsymbol{\sigma}_{\mathcal{M}} \circ \tau_{\mathcal{M}}) = \psi_1(\boldsymbol{\sigma}_{\mathcal{M}}) \circ \underbrace{\psi_1(\tau_{\mathcal{M}})}_{\stackrel{\text{def}}{=} \tau_{\mathcal{P}}} = \boldsymbol{\sigma}_{\mathcal{P}} \circ \tau_{\mathcal{P}} \tag{13a}$$

$$\psi_1(\boldsymbol{\pi}_{\mathcal{M}}) = \psi_1(\boldsymbol{\sigma}_{\mathcal{M}} \circ \tau_{\mathcal{M}}) = \psi_1(\boldsymbol{\sigma}_{\mathcal{M}}) \circ \underbrace{\psi_1(\tau_{\mathcal{M}})}_{\stackrel{\text{def}}{=} \tau_{\mathcal{P}} \circ \tau'_{\mathcal{P}}} = \boldsymbol{\sigma}_{\mathcal{P}} \circ \tau_{\mathcal{P}} \circ \tau'_{\mathcal{P}} \tag{13b}$$

---

[27]Note that we treat any transitions that have 0 weight as non-existing, and vice versa. A careful treatment could distinguish weight 0 and non-existence in the model.

$$(q_\iota, \sqcup) \xrightarrow{y_1/N} (q_1, \gamma_1) \to \cdots \to \underbrace{(q_2, \gamma_2) \xrightarrow{y_2/R} (q_3, \gamma_3) \to}_{\sigma_\mathcal{M}} \underbrace{(q_4, \gamma_4) \xrightarrow{y_3/L} (q_5, \gamma_5)}_{\tau_\mathcal{M}}$$

$$\underbrace{q_\iota \xrightarrow[\varepsilon \to \varepsilon]{y_1, \bot, \bot \to \gamma_2} q_1 \to \cdots \to q_2' \xrightarrow[\varepsilon \to \gamma_3]{y_2, \gamma_2, \gamma_2 \to \varepsilon} q_3 \to}_{\psi_1(\sigma_\mathcal{M})} q_4 \underbrace{\xrightarrow[\varepsilon \to \varepsilon]{y_4, \gamma_4, \gamma_4 \to \gamma_5} q_\tau' \xrightarrow[\gamma' \to \varepsilon]{\varepsilon, \gamma_4, \varepsilon \to \gamma'} q_5}_{\tau_\mathcal{P} \tau_{\mathcal{P}'}}$$

$$\underbrace{\qquad\qquad\qquad\qquad\qquad\qquad\qquad\qquad\qquad\qquad}_{\psi_1(\pi_\mathcal{M})}$$

Figure 5: Illustration of how $\psi_1$ maps a path in the original $\mathbb{Q}$PTM (top half) to a path in the 2PDA (bottom half). Note that those states with a prime do *not* correspond to states in the original $\mathbb{Q}$PTM.

depending on whether $\tau_\mathcal{M}$ corresponds to one (Eq. (13a)) or two transitions (Eq. (13b)) in $\mathcal{P}$. Note that the extension $\psi_1$ preserves the weight and yield of $\pi_\mathcal{M}$. This is clear by inspecting the construction as exactly one transition inherits $\tau_\mathcal{M}$'s weight and yield. In the two-transition case, the second transition is given weight 1 and the yield $\varepsilon$.

($\Longleftarrow$) Next, we define a weight- and yield-preserving mapping from the subpaths $\pi_\mathcal{P}$ in the constructed 2PDA to the subpaths $\pi_\mathcal{M}$ in the $\mathcal{M}$. Fix an arbitrary $Q$-subpath $\pi_\mathcal{P}$ in $\mathcal{P}$. Our proof proceeds by structural induction on the subpath relation: $\sigma_\mathcal{P} \prec \pi_\mathcal{P} \iff \sigma_\mathcal{P}$ is a strict $Q$-subpath of $\pi_\mathcal{P}$. Note that this is a well-founded ordering, making it suitable for induction.

**Inductive Hypothesis.** For all $\sigma_\mathcal{P} \prec \pi_\mathcal{P}$, the function $\psi_2$ is weight- and yield-preserving.

**Base Case.** The function $\psi_2$ is defined to map the empty subpath in $\mathcal{P}$ to the empty subpath in $\mathcal{M}$ with weight 1 and yield $\varepsilon$. This preserves the weight and yield.

**Inductive Case.** Let $\pi_\mathcal{P}$ be an arbitrary non-empty $Q$-subpath in $\mathcal{P}$. Consider the figure below:

$$\underbrace{q_\iota \xrightarrow[\varepsilon \to \varepsilon]{y_1, \bot, \bot \to \gamma_2} q_1 \to \cdots \to q_2 \xrightarrow[\varepsilon \to \gamma_3]{y_2, \gamma_2, \gamma_2 \to \varepsilon} q_3 \to}_{\sigma_\mathcal{P}} \underbrace{q_4 \xrightarrow[\varepsilon \to \varepsilon]{y_4, \gamma_4, \gamma_4 \to \gamma_5} q_\tau' \xrightarrow[\gamma' \to \varepsilon]{\varepsilon, \gamma_4, \varepsilon \to \gamma'} q_5}_{\tau_\mathcal{P} \tau_{\mathcal{P}'}}$$

$$\underbrace{\qquad\qquad\qquad\qquad\qquad\qquad\qquad}_{\pi_\mathcal{P}}$$

$$\underbrace{(q_\iota, \sqcup) \xrightarrow{y_1/N} (q_1, \gamma_1) \to \cdots \to (q_2, \gamma_2) \xrightarrow{y_2/R} (q_3, \gamma_3) \to}_{\psi_2(\sigma_\mathcal{P})} \underbrace{(q_4, \gamma_4) \xrightarrow{y_3/L} (q_5, \gamma_5)}_{\tau_\mathcal{M}}$$

$$\underbrace{\qquad\qquad\qquad\qquad\qquad\qquad\qquad}_{\psi_2(\pi_\mathcal{P})}$$

Figure 6: Illustration of how $\psi_2$ maps a path in the 2PDA (top half) to a path in the original $\mathbb{Q}$PTM (bottom half). Note that those states with a prime do *not* correspond to states in the original $\mathbb{Q}$PTM.

By construction, this path can be decomposed into a strict $Q$-subpath $\sigma_\mathcal{P} \prec \pi_\mathcal{P}$ and either one or two additional transitions as defined in (i)–(iii), as shown in Fig. 6. In the single-transition case, post-pended to $\sigma_\mathcal{P}$ is either a single transition in $\mathcal{M}$ (case (i) and case (ii)). In the two-transition case, post-pended to $\sigma_\mathcal{P}$ is a pair of the two consecutive transitions of $\mathcal{P}$ that together simulate a single transition in $\mathcal{M}$ (case (iii)), shown at the top of Fig. 6. In the first case, $\psi_2$ maps that action in $\pi_\mathcal{P}$ to its (unique) corresponding action in $\mathcal{M}$. In the latter case, note that due to the uniqueness of the added named state $q_\tau'$ for the transition of type (iii) in $\mathcal{M}$, all non-zero-weighted paths containing $q_\tau'$ in $\mathcal{P}$ will contain *both* transitions

in $\mathcal{P}$ defined in (iii), consecutively. Such pairs of transitions have a corresponding transition in the original $\mathbb{Q}$PTM—the transition for which they were added. Thus, we can extend $\psi_2$ in the following manner:

$$\psi_2(\boldsymbol{\pi}_\mathcal{P}) = \psi_2(\boldsymbol{\sigma}_\mathcal{P} \circ \tau_\mathcal{P}) = \psi_2(\boldsymbol{\sigma}_\mathcal{P}) \circ \underbrace{\psi_2(\tau_\mathcal{P})}_{\overset{\text{def}}{=} \tau_\mathcal{M}} = \boldsymbol{\sigma}_\mathcal{M} \circ \tau_\mathcal{M} \tag{14a}$$

$$\psi_2(\boldsymbol{\pi}_\mathcal{P}) = \psi_2(\boldsymbol{\sigma}_\mathcal{P} \circ \tau_\mathcal{P} \circ \tau'_\mathcal{P}) = \psi_2(\boldsymbol{\sigma}_\mathcal{P}) \circ \underbrace{\psi_2(\tau_\mathcal{P} \circ \tau'_\mathcal{P})}_{\overset{\text{def}}{=} \tau_\mathcal{M}} = \boldsymbol{\sigma}_\mathcal{M} \circ \tau_\mathcal{M} \tag{14b}$$

depending on whether $\tau_\mathcal{M}$ corresponds to one (Eq. (14a)) or two transitions (Eq. (14b)) in $\mathcal{P}$. Note that the extension $\psi_2$ preserves the weight and yield. This is clear by inspecting the construction as exactly one transition inherits $\tau_\mathcal{M}$'s weight and yield. In the two-transition case, the second transition is given weight 1 and the yield $\varepsilon$.

**Wrapping up.** Thus, we have defined a pair ($\psi_1$ and $\psi_2$) of weight-preserving, yield-preserving total functions that map arbitrary $Q$-subpaths[28] in $\mathcal{M}$ to paths in $\mathcal{P}$ and vice versa. It is easy to see that the two maps are inverses of each other; $\psi_2$ undoes the operations of $\psi_1$, and vice versa. This means that $\psi_1$ is a bijection. Finally, because all halting paths in $\mathcal{M}$ are $Q$-subpaths, we conclude $\mathcal{M}$ and $\mathcal{P}$ are strongly equivalent.

    ( $\Longleftarrow$ ) To prove the backward direction, we want to show that any 2PDA $\mathcal{P}$ has a strongly equivalent $\mathbb{Q}$PTM $\mathcal{M}$. We proceed analogously: Given a 2PDA $\mathcal{P}$, we construct a candidate $\mathbb{Q}$PTM $\mathcal{M}$ and then sketch a path level weight- and yield-preserving bijection, again in the form of two injective functions which are inverses of one another. This proves strong equivalence.

**Construction of $\mathcal{M}$.** Let $\mathcal{P} = (Q, \Sigma, \Gamma, \delta, q_\iota, q_\varphi)$ be an arbitrary probabilistic 2PDA. Now we define $\mathbb{Q}$PTM $\mathcal{M} = (Q_\mathcal{M}, \Sigma, \Gamma, \delta_\mathcal{M}, q_\iota, q_\varphi)$ to have the same alphabets $\Sigma, \Gamma$ as $\mathcal{P}$. Furthermore, we let $\mathcal{M}$ have a superset of the states of $\mathcal{P}$, that is, we let $Q_\mathcal{M} = Q \cup Q'$, where $Q'$ are some additional states. We define the transitions of $\mathcal{M}$ by enumerating and distinguishing between *all* the possible transition types in $\mathcal{P}$:

(i) **Transitions that do not pop or push any symbols**, i.e., $q \xrightarrow[\varepsilon \to \varepsilon]{y, \gamma, \varepsilon \to \varepsilon} q'$. For such transitions, we define the equally weighted stay-in-place operation ($N$) of the form $(q, \gamma) \xrightarrow{y/N} (q', \gamma)$ in $\mathcal{M}$.

(ii) **Transitions that pop a symbol and push a symbol to the same stack**, i.e., (a) $q \xrightarrow[\varepsilon \to \varepsilon]{y, \gamma, \gamma_1 \to \gamma_3} q'$ with $\gamma_1, \gamma_3 \neq \varepsilon$ or (b) $q \xrightarrow[\gamma_2 \to \gamma_4]{y, \gamma, \varepsilon \to \varepsilon} q'$ with $\gamma_2, \gamma_4 \neq \varepsilon$. Each transition of type (a) defines a single $\mathbb{Q}$PTM transition $(q, \gamma_1) \xrightarrow{y/N} (q', \gamma_3)$ with the same weight as the original transition. Each transition of type (b) defines two consecutive transitions $(q, \gamma) \xrightarrow{y/L} (q_\tau, \gamma), (q_\tau, \gamma_2) \xrightarrow{\varepsilon/R} (q', \gamma_4)$, where we create a new state $q_\tau$ unique to the given transition in $\mathcal{P}$. The weight of the first of the two new transitions in $\mathcal{P}$ equals the weight of the original transition in $\mathcal{M}$, and the second one has weight 1 (thus being the only possible continuation from state $q_\tau$).

(iii) **Transitions that pop from one stack and push to the other stack**, i.e., (a) $q \xrightarrow[\varepsilon \to \gamma_4]{y, \gamma, \gamma_1 \to \varepsilon} q'$ with $\gamma_1, \gamma_4 \neq \varepsilon$ or (b) $q \xrightarrow[\gamma_2 \to \varepsilon]{y, \gamma, \varepsilon \to \gamma_3} q'$ with $\gamma_2, \gamma_3 \neq \varepsilon$. A transition of type (a) defines a transition in a $\mathbb{Q}$PTM that moves the head to the right (weighted equally to the original transition): $(q, \gamma_1) \xrightarrow{y/R} (q', \gamma_4)$. A transition of type (b) defines a $\mathbb{Q}$PTM transition moving to the left followed by a stay-in-place operation that changes the symbol below the head: $(q, \gamma) \xrightarrow{y/L} (q_\tau, \gamma), (q_\tau, \gamma_2) \xrightarrow{\varepsilon/N} (q', \gamma_3)$. Again, the weight of the first new transitions in $\mathcal{P}$ is chosen to be equal to the weight of the original transition in $\mathcal{M}$ and the latter has weight 1.

---

[28]All paths in $\mathcal{M}$ are $Q$-subpaths by definition.

(iv) **Transitions that push a symbol without popping one.** These can be thought of as insertions that move all the symbols on the right side of the head to the right. Thus, they can be simulated by *sequences* of actions by the $\mathbb{Q}$PTM. For instance, a transition of the form $q \xrightarrow[\gamma_3 \to \varepsilon]{y, \gamma, \varepsilon \to \varepsilon} q'$ where $\gamma_3 \neq \varepsilon$ can be simulated in $\mathcal{M}$ as follows:

    1.) Change the current symbol under the head ($\gamma$) to a marker symbol not in the alphabet, e.g., $\downarrow$;

    2.) Go to the end of the string and shift all the characters up to the marker one by one to the right, until back at the original position.

    3.) Replace $\downarrow$ with the symbol to be inserted ($\gamma_3$).

Therefore, the sequence of such transitions is added to $\mathcal{M}$ for any transition of this form in the 2PDA. We preserve the weight of the original transition by defining the weight of the first transition in step 1.) to be the weight of the original transition in $\mathcal{P}$ and the weights of all following transitions in steps 2.) and 3.) to be 1.

(v) **Transitions that pop from one or both stacks without pushing equally many symbols.** These can be thought of as deletions that remove symbols from the $\mathcal{M}$'s tape and move all the symbols on the right of the head to the left. For instance, a transition $q \xrightarrow[\varepsilon \to \varepsilon]{y, \gamma, \gamma_1 \to \varepsilon} q'$ where $\gamma_1 \neq \varepsilon$ can be modeled using the strategy from (iv), where step 2.) changes to shifting all the symbols to the left one by one until reaching the end of the string, then moving back to the marker. We define the sequence of such transitions in $\mathcal{M}$ for any transition of this form in the 2PDA. Again, the first transition has the same weight as the original transition in $\mathcal{P}$ and all following new transitions have weight 1.

(vi) **Transitions that pop symbols from both stacks and push to both stacks**, i.e., $q \xrightarrow[\gamma_2 \to \gamma_4]{y, \gamma, \gamma \to \gamma_3} q'$, where $\gamma_1, \gamma_2, \gamma_3, \gamma_4 \neq \varepsilon$. Such transitions can be regarded as a composition of the two cases of (ii), performing first the simulation from (b) and then from (a). The chaining can be done by adding another intermediate state, $q_{\tau'}$. Such transitions in the 2PDA therefore define the sequence of transitions $(q, \gamma) \xrightarrow{y/L} (q_\tau, \gamma), (q_\tau, \gamma_2) \xrightarrow{\varepsilon/R} (q_{\tau'}, \gamma_4), (q_{\tau'}, \gamma) \xrightarrow{\varepsilon/N} (q', \gamma_3)$ in the $\mathbb{Q}$PTM. The weights are again chosen such that the weight of the first one corresponds to that of the original transition in $\mathcal{P}$, and all following ones have weight 1.[29]

**A weight- and yield-preserving bijection.** With the construction above, we again show that there is a weight- and yield-preserving bijection between halting paths of $\mathcal{M}$ and $\mathcal{P}$. The reasoning is analogous to the one in the forward direction. While we are only interested in halting (accepting) paths, i.e., paths going from $q_\iota$ to $q_\varphi$ with a non-zero weight, we again prove a stronger result that there exists a weight- and yield-preserving bijection between all $Q$-subpaths.

**Definition D.2.** *A $Q$-subpath in $\mathcal{M}$ is a subpath whose last state corresponds to a state that also exists in the original $\mathbb{Q}$PTM. Note that $\mathcal{P}$'s states are constructed to be a sub*set *of those in $\mathcal{M}$.*

($\Rightarrow$) We define a weight- and yield-preserving mapping $\psi_3$ from the subpaths $\pi_\mathcal{P}$ in the original $\mathcal{P}$ to the subpaths $\pi_\mathcal{M}$ in the constructed $\mathcal{M}$. Fix an arbitrary $Q$-subpath $\pi_\mathcal{P}$ in $\mathcal{P}$. Our proof proceeds by structural induction on the subpath relation: $\sigma_\mathcal{P} \prec \pi_\mathcal{P} \iff \sigma_\mathcal{P}$ is a strict $Q$-subpath of $\pi_\mathcal{P}$. Note that this is a well-founded ordering, making it suitable for induction.

**Inductive Hypothesis.** For all $\sigma_\mathcal{P} \prec \pi_\mathcal{P}$, the function $\psi_3$ is weight- and yield-preserving.

**Base Case.** The function $\psi_3$ is defined to map the empty subpath to itself with weight 1 and yield $\varepsilon$. This preserves the weight and yield.

---

[29]Note that the same kind of composition can be used to achieve popping or pushing more than one symbol in (iv) and (v).

$$q_\iota \xrightarrow[\varepsilon \to \varepsilon]{y_1, \bot, \bot \to \gamma_1} q_1 \to \cdots \to q_2 \xrightarrow[\varepsilon \to \gamma_3]{y_2, \gamma_2, \gamma_2 \to \varepsilon} q_3 \to \cdots \to q_4 \xrightarrow[\gamma_5 \to \gamma_6]{y_3, \gamma_4, \varepsilon \to \varepsilon} q_5$$

$$\underbrace{\hspace{5cm}}_{\boldsymbol{\sigma}_{\mathcal{P}}} \quad \underbrace{\hspace{2cm}}_{\tau_{\mathcal{P}}}$$

$$\underbrace{\hspace{8cm}}_{\boldsymbol{\pi}_{\mathcal{P}}}$$

$$(q_\iota, \sqcup) \xrightarrow{y_1/N} (q_1, \gamma_1) \to \ldots \to (q_2, \gamma_2) \xrightarrow{y_2/R} (q_3, \gamma_3) \to \ldots \to (q_4, \gamma_4) \xrightarrow{y_3/L} (q'_\tau, \gamma_4), (q'_\tau, \gamma_5) \xrightarrow{\varepsilon/R} (q_5, \gamma_6)$$

$$\underbrace{\hspace{5cm}}_{\psi_3(\boldsymbol{\sigma}_{\mathcal{P}})} \quad \underbrace{\hspace{4cm}}_{\tau_{\mathcal{M}} \tau_{\mathcal{M}'}}$$

$$\underbrace{\hspace{9cm}}_{\psi_3(\boldsymbol{\pi}_{\mathcal{P}})}$$

Figure 7: Illustration of how $\psi_3$ maps a path in the original 2PDA (top half) to a path in the $\mathbb{Q}$PTM (bottom half). Note that those states with a prime do *not* correspond to states in the original 2PDA.

**Inductive Case.** Let $\boldsymbol{\pi}_{\mathcal{P}}$ be an arbitrary non-empty subpath in $\mathcal{P}$. As depicted in Fig. 7, let $\boldsymbol{\sigma}_{\mathcal{P}}$ be the (strict) subpath of $\boldsymbol{\pi}_{\mathcal{P}}$ that omits the last transition $\tau_{\mathcal{P}}$. Then, $\boldsymbol{\sigma}_{\mathcal{P}}$ is mapped as follows: $\boldsymbol{\sigma}_{\mathcal{P}} \mapsto \psi_3(\boldsymbol{\sigma}_{\mathcal{P}})$. This mapping exists and is weight- and yield-preserving by the inductive hypothesis. Now, we seek to extend the function $\psi_3$ to $\boldsymbol{\pi}_{\mathcal{P}} = \boldsymbol{\sigma}_{\mathcal{P}} \circ \tau_{\mathcal{P}}$. By our construction of $\mathcal{M}$, each of the transitions in $\boldsymbol{\pi}_{\mathcal{P}}$ is simulated by a number of transitions in $\mathcal{M}$. Importantly, the transitions corresponding to any transition $\tau_{\mathcal{P}}$ in $\mathcal{P}$ are *unique* to the particular $\tau_{\mathcal{P}}$ due to the $\mathcal{P}$-transition-dependant naming of the added states $Q'$ in $\mathcal{M}$. Thus, we can extend $\psi_3$ in the following manner:

$$\psi_3(\boldsymbol{\pi}_{\mathcal{P}}) = \psi_3(\boldsymbol{\sigma}_{\mathcal{P}} \circ \tau_{\mathcal{P}}) = \psi_3(\boldsymbol{\sigma}_{\mathcal{P}}) \circ \underbrace{\psi_3(\tau_{\mathcal{P}})}_{\overset{\text{def}}{=} \tau_{\mathcal{M}}} = \boldsymbol{\sigma}_{\mathcal{M}} \circ \tau_{\mathcal{M}} \tag{15a}$$

$$\psi_3(\boldsymbol{\pi}_{\mathcal{P}}) = \psi_3(\boldsymbol{\sigma}_{\mathcal{P}} \circ \tau_{\mathcal{P}}) = \psi_3(\boldsymbol{\sigma}_{\mathcal{P}}) \circ \underbrace{\psi_3(\tau_{\mathcal{P}})}_{\overset{\text{def}}{=} \tau_{\mathcal{M},1} \circ \cdots \circ \tau_{\mathcal{M}, L_{\tau_{\mathcal{P}}}}} = \boldsymbol{\sigma}_{\mathcal{M}} \circ \tau_{\mathcal{M},1} \circ \cdots \circ \tau_{\mathcal{M}, L_{\tau_{\mathcal{P}}}} \tag{15b}$$

where $\tau_{\mathcal{M},1}, \ldots, \tau_{\mathcal{M}, L_{\tau_{\mathcal{P}}}}$ are the $L_{\tau_{\mathcal{P}}}$ transitions in $\mathcal{M}$ that the transition $\tau_{\mathcal{P}}$ corresponds to.[30] Note that the extension $\psi_3$ preserves the weight and yield of $\boldsymbol{\pi}_{\mathcal{P}}$. This is clear by inspecting the construction as exactly one transition inherits $\tau_{\mathcal{M}}$'s weight and yield, while the remaining transitions have weight 1 and the yield $\varepsilon$.

($\Leftarrow$) Next, we define a weight- and yield-preserving mapping from the subpaths $\boldsymbol{\pi}_{\mathcal{M}}$ in the constructed $\mathbb{Q}$PTM to the subpaths $\boldsymbol{\pi}_{\mathcal{P}}$ in the $\mathcal{P}$. Fix an arbitrary $Q$-subpath $\boldsymbol{\pi}_{\mathcal{M}}$ in $\mathcal{M}$. Our proof proceeds by structural induction on the subpath relation: $\boldsymbol{\sigma}_{\mathcal{M}} \prec \boldsymbol{\pi}_{\mathcal{M}} \iff \boldsymbol{\sigma}_{\mathcal{M}}$ is a strict $Q$-subpath of $\boldsymbol{\pi}_{\mathcal{M}}$. Note that this is a well-founded ordering, making it suitable for induction.

**Inductive Hypothesis.** For all $\boldsymbol{\sigma}_{\mathcal{M}} \prec \boldsymbol{\pi}_{\mathcal{M}}$, the function $\psi_4$ is weight- and yield-preserving.

**Base Case.** The function $\psi_4$ is defined to map the empty subpath in $\mathcal{M}$ to the empty subpath in $\mathcal{P}$ with weight 1 and yield $\varepsilon$. This preserves the weight and yield.

**Inductive Case.** Let $\boldsymbol{\pi}_{\mathcal{M}}$ be an arbitrary non-empty $Q$-subpath in $\mathcal{M}$. Consider the figure below:

By construction, this path can be decomposed into a strict $Q$-subpath $\boldsymbol{\sigma}_{\mathcal{M}} \prec \boldsymbol{\pi}_{\mathcal{M}}$ and either one or two additional transitions as defined in (i)–(vi), as shown in Fig. 8. In the single-transition case, post-pended to $\boldsymbol{\sigma}_{\mathcal{M}}$ is either a single transition in $\mathcal{M}$ (type (iia) and type (iiia)). In the multi-transition case, post-pended to $\boldsymbol{\sigma}_{\mathcal{M}}$ is a sequence of consecutive transitions of $\mathcal{M}$ that together simulate a single transition in $\mathcal{P}$, e.g. two consecutive new transitions added for a $\mathcal{P}$-transition of type (iib) as shown in the example at the top of Fig. 6. In the first case, $\psi_2$ maps that action in $\boldsymbol{\pi}_{\mathcal{M}}$ to its (unique) corresponding action in $\mathcal{P}$. In the case of multiple transitions, note that due to the uniqueness of the added named states in $\mathcal{M}$ ($q'_\tau$ for the transition of type (iib) in the example), all non-zero-weighted paths containing such transition-specific additional

---

[30]In cases (iib) and (iiib), $L_{\tau_{\mathcal{P}}} = 2$, and in case (vi) $L_{\tau_{\mathcal{P}}} = 3$. In cases (iv) and (v), $L_{\tau_{\mathcal{P}}}$ is linearly bounded by the number of symbols printed on the tape of $\mathcal{M}$ to the right of its head at the start of simulating $\mathcal{P}$'s transition.

$$(q_\iota, \sqcup) \xrightarrow{y_1/N} (q_1, \gamma_1) \rightarrow \ldots \rightarrow (q_2, \gamma_2) \xrightarrow{y_2/R} (q_3, \gamma_3) \rightarrow \ldots \rightarrow \underbrace{(q_4, \gamma_4) \xrightarrow{y_3/L} (q'_\tau, \gamma_4), (q'_\tau, \gamma_5)}_{\tau_\mathcal{M}\tau_\mathcal{M'}} \xrightarrow{\varepsilon/R} (q_5, \gamma_6)$$

$$\underbrace{(q_\iota, \sqcup) \xrightarrow{y_1/N} (q_1, \gamma_1) \rightarrow \ldots \rightarrow (q_2, \gamma_2)}_{\boldsymbol{\sigma}_\mathcal{M}}$$

$$\underbrace{\phantom{(q_\iota, \sqcup) \xrightarrow{y_1/N} (q_1, \gamma_1) \rightarrow \ldots \rightarrow (q_2, \gamma_2) \xrightarrow{y_2/R} (q_3, \gamma_3) \rightarrow \ldots \rightarrow (q_4, \gamma_4) \xrightarrow{y_3/L} (q'_\tau, \gamma_4), (q'_\tau, \gamma_5) \xrightarrow{\varepsilon/R} (q_5, \gamma_6)}}_{\boldsymbol{\pi}_\mathcal{M}}$$

$$q_\iota \xrightarrow[\varepsilon \rightarrow \varepsilon]{y_1, \perp, \perp \rightarrow \gamma_1} q_1 \rightarrow \cdots \rightarrow q_2 \xrightarrow[\varepsilon \rightarrow \gamma_3]{y_2, \gamma_2, \gamma_2 \rightarrow \varepsilon} q_3 \rightarrow \cdots \rightarrow q_4 \underbrace{\xrightarrow[\gamma_5 \rightarrow \gamma_6]{y_3, \gamma_4, \varepsilon \rightarrow \varepsilon} q_5}_{\tau_\mathcal{P}}$$

$$\underbrace{q_\iota \xrightarrow[\varepsilon \rightarrow \varepsilon]{y_1, \perp, \perp \rightarrow \gamma_1} q_1 \rightarrow \cdots \rightarrow q_2 \xrightarrow[\varepsilon \rightarrow \gamma_3]{y_2, \gamma_2, \gamma_2 \rightarrow \varepsilon} q_3 \rightarrow \cdots \rightarrow q_4}_{\psi_4(\boldsymbol{\sigma}_\mathcal{M})}$$

$$\underbrace{\phantom{q_\iota \xrightarrow{} q_1 \rightarrow \cdots \rightarrow q_2 \xrightarrow{} q_3 \rightarrow \cdots \rightarrow q_4 \xrightarrow{} q_5}}_{\psi_4(\boldsymbol{\pi}_\mathcal{M})}$$

Figure 8: Illustration of how $\psi_4$ maps a path in the original $\mathbb{Q}$PTM (top half) to a path in the 2PDA (bottom half). Note that those states with a prime do *not* correspond to states in the original 2PDA.

states in $\mathcal{M}$ will contain *all* the transitions in $\mathcal{M}$ that belong its sequence, consecutively. Such sequences of transitions have a corresponding transition in the original $\mathcal{P}$—the transition for which they were added.

Thus, we can extend $\psi_4$ in the following manner:

$$\psi_4(\boldsymbol{\pi}_\mathcal{M}) = \psi_4(\boldsymbol{\sigma}_\mathcal{M} \circ \tau_\mathcal{M}) = \psi_4(\boldsymbol{\sigma}_\mathcal{M}) \circ \underbrace{\psi_4(\tau_\mathcal{M})}_{\stackrel{\text{def}}{=} \tau_\mathcal{P}} = \boldsymbol{\sigma}_\mathcal{P} \circ \tau_\mathcal{P} \tag{16a}$$

$$\psi_4(\boldsymbol{\pi}_\mathcal{M}) = \psi_4(\boldsymbol{\sigma}_\mathcal{M} \circ \tau_{\mathcal{M},1} \circ \cdots \circ \tau_{\mathcal{M}, L_{\tau_\mathcal{P}}}) = \psi_4(\boldsymbol{\sigma}_\mathcal{M}) \circ \underbrace{\psi_4(\tau_{\mathcal{M},1} \circ \cdots \circ \tau_{\mathcal{M}, L_{\tau_\mathcal{P}}})}_{\stackrel{\text{def}}{=} \tau_\mathcal{P}} = \boldsymbol{\sigma}_\mathcal{P} \circ \tau_\mathcal{P} \tag{16b}$$

depending on whether $\tau_\mathcal{P}$ corresponds to one (Eq. (16a)) or multiple transitions (Eq. (16b)) in $\mathcal{M}$. Here again $\tau_{\mathcal{M},1}, \ldots, \tau_{\mathcal{M}, L_{\tau_\mathcal{P}}}$ are the $L_{\tau_\mathcal{P}}$ transitions in $\mathcal{M}$ the transition $\tau_\mathcal{P}$ corresponds to. Note that the extension $\psi_4$ preserves the weight and yield. This is clear by inspecting the construction as exactly one transition inherits $\tau_\mathcal{P}$'s weight and yield. In the multi-transition case, the second transition is given weight 1 and the yield $\varepsilon$.

**Wrapping up.** Thus, we have defined a pair ($\psi_3$ and $\psi_4$) of weight-preserving, yield-preserving total functions that map arbitrary $Q$-subpaths[31] in $\mathcal{P}$ to paths in $\mathcal{M}$ and vice versa. It is easy to see that the two maps are inverses of each other; $\psi_3$ undoes the operations of $\psi_4$, and vice versa. This means that $\psi_3$ is a bijection. Finally, because all halting paths in $\mathcal{P}$ are $Q$-subpaths, we conclude $\mathcal{P}$ and $\mathcal{M}$ are strongly equivalent. We therefore conclude that the classes of 2PDA and $\mathbb{Q}$PTM are strongly equivalent. ∎

## E  Proof of Thm. 3.2

The following theorem formalizes the informal claim made by Thm. 3.2, which says that every $\Sigma$-deterministic probabilistic 2PDAs admits a strongly equivalent $\varepsilon$RLMs, establishing a lower bound on the expressivity of $\varepsilon$RLMs. Due to the $\Sigma$-determinism, the proof is a simple extension of the weighted extension to Minsky's construction recently detailed in Svete and Cotterell (2023)—it uses the correspondence between the paths in the 2PDA and the $\varepsilon$RLM produced by Chung and Siegelmann's (2021) construction to define a natural weighting of the paths resulting in a weight- and yield-preserving mapping.

**Theorem E.1.** *Let $p$ be a language model defined by the $\Sigma$-deterministic probabilistic 2PDA $\mathcal{P}$. Let $\mathcal{R}$ be an RNN over $\overline{\Sigma}_\varepsilon$ as defined by Chung and Siegelmann's (2021) construction that furthermore defines*

---
[31]All paths in $\mathcal{P}$ are $Q$-subpaths by definition.

the $\varepsilon$RLM $p_{\mathcal{R}}$ with the output matrix $\mathbf{E} \in \mathbb{Q}^{|\overline{\Sigma}_\varepsilon| \times D}$:

$$E_{y,(\gamma,q)} \overset{\text{def}}{=} p\left(y \mid \gamma, q\right) \qquad \text{for } y \in \Sigma_\varepsilon \tag{17}$$

$$E_{\text{EOS},(\gamma,q)} \overset{\text{def}}{=} \begin{cases} 1 & \textbf{if } q = q_\varphi \\ 0 & \textbf{if } q \neq q_\varphi \end{cases}. \tag{18}$$

and the projection function $\boldsymbol{\pi}\left(\mathbf{x}\right) = \text{sparsemax}\left(\mathbf{x}\right) = \text{argmin}_{\mathbf{z} \in \boldsymbol{\Delta}^{|\overline{\Sigma}_\varepsilon|-1}} \|\mathbf{x} - \mathbf{z}\|_2$. Then, it holds that $p_{\mathcal{R}}$ is strongly equivalent to $p$.

*Proof.* We construct a weight- and yield-preserving bijection between the paths in 2PDA and $\mathcal{R}$. Let $\boldsymbol{\pi}_{\mathcal{P}}$ be a path in $\mathcal{P}$:

$$\boldsymbol{\pi}_{\mathcal{P}} = q_\iota \xrightarrow[\gamma_0^2 \to \gamma_1^2]{y_1, \perp, \perp \to \gamma_1^1} q_1 \to \cdots \to q_{N-1} \xrightarrow[\gamma_{N-1}^2 \to \gamma_N^2]{y_N, \gamma_{N-1}^1, \gamma_{N-1}^1 \to \gamma_N^1} q_\varphi$$

The definition of the RNN controller based on the construction by Chung and Siegelmann (2021) ensures that the hidden states $\mathbf{h}_t$ of the RNN are correctly updated, that is, that there is a one-to-one mapping between the hidden states of the RNN and the configurations of the 2PDA at every step of the construction. This ensures that $\overline{\mathbf{h}}_t = [\![\gamma_{1,t}, q_t]\!]$ for every $t$. Let $\psi$ be the function mapping paths in the 2PDA to those in $\mathcal{R}$ as

$$\psi\left(\boldsymbol{\pi}_{\mathcal{P}}\right) \overset{\text{def}}{=} \mathbf{h}_0 \circ \mathbf{h}_1 \circ \cdots \circ \mathbf{h}_N. \tag{19}$$

Due to the correctness of the RNN transition function, $\psi$ is yield-preserving. Moreover, the mapping is bijective—the one-to-one correspondence between the hidden states and the configurations of the 2PDA ensures that there is a one-to-one correspondence between the 2PDA transitions and the transitions between the hidden states, which furthermore means that the expansion of the mapping to entire paths is bijective.

Furthermore, the form of the hidden states $\mathbf{h}$ of the RNN enables the simple transformation of $\mathbf{h}_t$ into $\overline{\mathbf{h}}_t = [\![\gamma_{1,t}, q_t]\!]$ for every $t$, where $\gamma_{1,t}$ is the top symbol of stack 1 and $q_t$ the state of $\mathcal{P}$ at time step $t$. Notice that by the definition of $\Sigma$-determinism, $\overline{\mathbf{h}}_t$ contains all the information we need to determine the transition probabilities of $\mathcal{P}$ given any input symbol $y \in \Sigma_\varepsilon$. More precisely, it holds that for all $y \in \Sigma_\varepsilon$

$$\text{sparsemax}\left(\mathbf{E}\overline{\mathbf{h}}_t\right)_y = p\left(y \mid \gamma, q\right) \tag{20}$$

Lastly, the definition of $E_{\text{EOS},(\gamma,q)}$ ensures that $\mathcal{R}$ outputs EOS if and only if the simulated 2PDA is in its final state, $q_\varphi$.

From Eq. (20), it is easy to see that the probability of $\boldsymbol{\pi}_{\mathcal{P}}$ is

$$p\left(\boldsymbol{\pi}_{\mathcal{P}}\right) = \prod_{t=1}^{N} p\left(y_t \mid \gamma_{t-1}^1, q_{t-1}\right) = \prod_{t=1}^{N} p\left(y_t \mid \gamma_{t-1}^1, q_{t-1}\right) = p_{\mathcal{R}}\left(\psi\left(\boldsymbol{\pi}_{\mathcal{P}}\right)\right), \tag{21}$$

showing that $\psi$ is indeed weight-preserving, which finishes the proof. $\blacksquare$

# F    Comparison Between Different Variants of Probabilistic Turing Machines

We have introduced a number of different formulations of PTMs (and 2PDA) with various additions and restrictions in the course of this work. In this section of the appendix, we compare the types of distributions that can be expressed by each of these variants of PTMs. For an illustrative overview of the distributions that can be expressed by the different models, see Fig. 9.

**Definition F.1.** *A PTM is **deterministic** if, for any configuration $q, \gamma \in Q \times \Gamma$, and any symbol $y \in \Sigma_\varepsilon$, there is at most one transition starting at that configuration and emitting $y$. Furthermore, if there is a transition starting in $(q, \gamma)$ outputting $\varepsilon$, both transitions must be identical.*

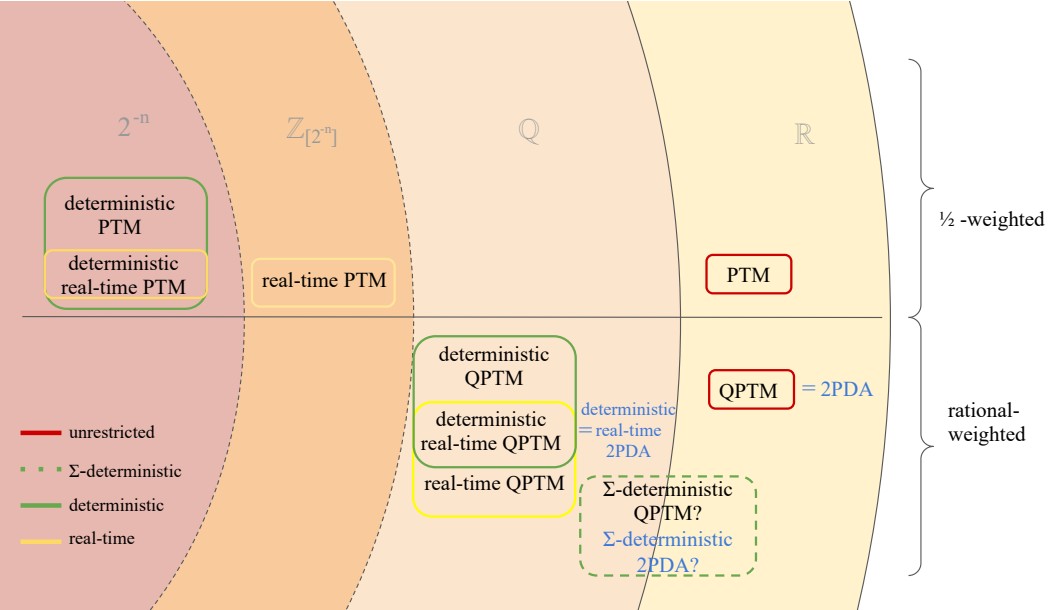

Figure 9: A schematic illustration of the different types of Turing machines and 2PDA and their corresponding place in a hierarchy of distributions. The curves differentiate different types of distributions, where $2^{-n}$ means every string has a binary probability, $\mathbb{Z}_{[2^{-n}]}$ refers to the dyadic distributions, and $\mathbb{Q}$ and $\mathbb{R}$ are the rational-valued and real-valued distributions, respectively. The colors of the boxes indicate which restrictions are placed on the automata, from real-time and deterministic, via $\Sigma$-deterministic, to unrestricted. Finally, the horizontal line divides formulations of machines that have just two transition functions that are uniformly distributed vs. the case of finitely many rational-values transition functions. Note that the $\Sigma$-deterministic automata are placed between the rational-valued and the real-valued distributions.

Note that the definition above is more restrictive than that of a general PTM, but a superset of the class of deterministic Turing machines, $\mathcal{M}$s, which can be thought of as unweighted PTMs with the restriction that both transition functions are identical. It is a well-known result that $\mathcal{M}$s are computationally equivalent to PTMs, that is, they can recognize the same (unweighted) languages. However, in contrast to $\mathcal{M}$s, deterministic PTMs as defined above can still express a semimeasure over strings, albeit a trivial one (each string $\boldsymbol{y}$ that can be generated has a probability of $2^{-|\boldsymbol{y}|}$ of being generated).

**Proposition F.1.** *A deterministic* PTM *can only express distributions where each finite string has a binary probability, that is, a probability of the form* $2^{-n}$.

*Proof.* Let $\boldsymbol{y} \in \Sigma^*$ be a string. Because PTM is deterministic, there is a unique path in PTM that accepts $\boldsymbol{y}$. Let $n$ be the length of the accepting path. Then, $\mathbb{P}_{\mathcal{M}}(\boldsymbol{y}) = 2^{-n}$. ∎

**Definition F.2.** *A* PTM *is **real-time** if it has no $\varepsilon$-transitions.*

**Definition F.3.** *A **dyadic rational** is a rational number whose numerator can be any integer and whose denominator is a power of 2. We denote the set of dyadic rationals with* $\mathbb{Z}_{[2^{-n}]}$.[32]

**Proposition F.2.** *A real-time* PTM *can express only dyadic measures, that is, the measure of each string is a dyadic rational.*

*Proof.* Let $\boldsymbol{y} \in \Sigma^*$ be a string, and let $n = |\boldsymbol{y}|$. Because PTM is real-time, every path accepting $\boldsymbol{y}$ has length $n$. Moreover, since PTM is real-time, it may only have a finite number of accepting paths for any string. If PTM has $k$ accepting paths for the string $\boldsymbol{y}$, thne $\mathbb{P}_{\mathcal{M}}(\boldsymbol{y}) = k2^{-n}$, which is a dyadic rational. ∎

**A note on the language expressivity of real-time Turing machines.** While deterministic and non-deterministic Turing machines can recognize or generate the same languages, it has been shown that all real-time languages are context-sensitive (Burkhard and Varaiya, 1971). In fact, there are real-time

---

[32]Note that the binary rationals are a special case of dyadic rationals where the numerator has to be 1.

definable languages that are context-sensitive and not context-free. On the other hand, there are also context-free languages that are not real-time definable (Rosenberg, 1967). Real-time Turing machines with just one tape have been shown to only recognize regular languages (Tadaki et al., 2010). However, with just one more tape, the computational power increases dramatically (Rabin, 1963), allowing recognition of languages that are non-context-free (Rosenberg, 1967).

We now turn to the investigation of rationally weighted PTMs, i.e., $\mathbb{Q}$PTMs. First, recall that an unrestricted $\mathbb{Q}$PTM is weakly equivalent to an unrestricted PTM (Prop. 3.1). Furthermore, Icard (2020, Thm. 3) showed that PTMs define exactly the enumerable semimeasures (see Appendix G for details). This means that any enumerable real-valued semi-measure over strings can be expressed by a PTM, and, hence, a $\mathbb{Q}$PTM.

**Proposition F.3.** *Real-time deterministic $\mathbb{Q}$PTMs are strictly less expressive than general $\mathbb{Q}$PTMs.*

*Proof.* By Thm. G.1, PTMs, and hence $\mathbb{Q}$PTMs, can express real-valued semimeasures over strings. For instance, there exists a $\mathbb{Q}$PTM that generates the language $\Sigma^*$ for the one-symbol alphabet $\Sigma = \{a\}$, such that the probability of a string of a certain length is given by a Poisson measure: $p(a^k) = \mathrm{Pois}(\lambda, k) = \frac{\lambda^k e^{-\lambda}}{k!}$ for some $\lambda \in \mathbb{R}^+$. Let us choose, e.g., $\lambda = 1$. Then the probability of each string in $\Sigma^*$ is an irrational number. However, an RD–$\mathbb{Q}$PTM has to output a symbol at each time step with a rational probability, and hence, there exists no RD–$\mathbb{Q}$PTM that can express the above language. ∎

Note that similar arguments can be made to show that both determinism as well as real-time on their own are enough to restrict distributions from such $\mathbb{Q}$PTMs to the rationals.

**Corollary F.1.** *Deterministic 2PDA are strictly less expressive than non-deterministic 2PDA.*

*Proof.* This follows from the strong equivalence of $\mathbb{Q}$PTMs and 2PDA described in Thm. 3.1. ∎

Finally, we leave the case of $\Sigma$-deterministic probabilistic automata for future work, but we hypothesize that it lies in the realm of real-valued (including irrational) distributions.

## G   Every $\varepsilon$RLMs Has a Weakly Equivalent 2PDA

In this section, we re-state basic definitions and theorems from computable analysis and then use them to show that the computational expressivity of $\varepsilon$RLMs is bounded by that of a 2PDA.

**Definition G.1.** *Icard (2020, Ch. 3) A real number $r$ is **lower semi-computable** if there exists a sequence of computably enumerable rationals $\{q_n\}_{n=1}^{\infty}, q_n \in \mathbb{Q}$ that is i) monotonically increasing in $n$ and ii) converges to $r$ from below, i.e., $\lim_{n \to \infty} q_n = r$.*

**Definition G.2.** *A semimeasure $\mu$ over $\Sigma^*$ is called **enumerable** if for all $\boldsymbol{y} \in \Sigma^*$ we have $\mu(\boldsymbol{y}) = r$ for some $r$ that is lower semi-computable.*

**Theorem G.1.** *Icard (2020, Thm. 3) Probabilistic Turing machines define exactly the enumerable semimeasures.*

**Proposition 3.2.** *Every $\varepsilon$RLM has a weakly equivalent 2PDA.*

*Proof.* We first show that every $\varepsilon$RLM defines an enumerable semimeasure. Let $\mathcal{R}$ be an $\varepsilon$RLM defined over alphabet $\Sigma$. We seek to show that the measure of any string $\boldsymbol{y} \in \Sigma^*$ computed by $\mathcal{R}$ is lower semi-computable. In general, there are (countably) many runs that may generate a string $\boldsymbol{y}$. The measure of $\boldsymbol{y}$ in $\mathcal{R}$ is the sum of all the weights of all such runs. More formally, let $\{\widehat{\boldsymbol{y}}_i\}_{i=1}^{\infty} \subset \Sigma_\varepsilon^*$ be an enumeration of all the runs in $\mathcal{R}$ that generate the string $\boldsymbol{y} \in \Sigma^*$, i.e., for each $\widehat{\boldsymbol{y}}_i$, the yield of $\widehat{\boldsymbol{y}}_i$ is $\boldsymbol{y}$. Construct the following sequence

$$r_n = \sum_{i=1}^{n} p(\widehat{\boldsymbol{y}}_i) \tag{22}$$

Thus, $\lim_{n \to \infty} r_n = p(\boldsymbol{y})$ and $r_n$ is monotonically increasing in $n$. Furthermore, because $\mathcal{R}$ is rationally weighted, every $r_n$ is rational. Next, note by Prop. 3.1 and Thm. 3.1, the classes of PTM and 2PDA are

weakly equivalent. Then, combining this with Thm. G.1, we have that 2PDA can express any enumerable semimeasure. This proves the claim.

∎