# OpenReview forum: "On the Representational Capacity of Recurrent Neural Language Models"
_EMNLP/2023/Conference — EMNLP 2023 Main_

### Official Review · Reviewer_atVK · 2023-07-28

**Soundness:** 4

**Excitement:**

4: Strong: This paper deepens the understanding of some phenomenon or lowers the barriers to an existing research direction.

**Paper Topic And Main Contributions:**

This theoretical paper revisits the RNN expressiveness issues and make two contributions:
- Upperbound: the paper extends the results of Siegelmann and Sontag (which shows that RNN with unbounded computation time are turing complete) that an RNN with unbounded computation time can simulate any distribution specified by a turning machine. Technically, this involves handling the output matrix so that it projects to the probability space that the corresponding Turing machine specifies.
- Lowerbound: the paper insightfully notices that the unbounded computational time hypothesis is violated in practice where only linear computation time machines are used. Based on this observation, the paper uses similar techniques to upperbound proof to derive a lowerbound for linear RNN expressiveness and demonstrate the realtime deterministic Q probabilitic turing machine as a lower bound.

**Questions For The Authors:**

Do authors have some insights what direction should be worked on for tighter bound/better characterization?

**Reasons To Accept:**

The missings in the theory of the field makes theoretical contribution more important than before. Apart from the solid contribution that the paper has made including the upperbound and lowerbound of the RNNs, the lowerbound derivation actually comes from the observation of practical RNN usage, which I appreciate.

The paper is easily accessible by all NLP researchers, even people not very familiar with the formal language theory.

**Reasons To Reject:**

The paper doesn't introduce new proving techniques to give tighter bounds or offer further insights. The upperbound proof is an extension from Siegelmann and Sontag and the lowerbound proof follows the upperbound proving techniques.

**Reproducibility:**

4: Could mostly reproduce the results, but there may be some variation because of sample variance or minor variations in their interpretation of the protocol or method.

**Reviewer Confidence:**

3: Pretty sure, but there's a chance I missed something. Although I have a good feel for this area in general, I did not carefully check the paper's details, e.g., the math, experimental design, or novelty.

---

> ### Author Rebuttal · Authors · 2023-08-29
>
> Thank you for the time to read our work and for your review.
> We acknowledge that both the upper and lower bounds are shown using a similar technique to Siegelmann and Sontag. We think that this gives new insights specifically because it relates known theoretical results about general recurrent networks to currently used language models that use a similar, albeit different architecture, in that we now involve a step of sampling outputs that is not present in the original formulation.
> To answer your question, we believe one potential avenue for arriving at a tighter bound is to change the infinite precision assumption for one that allows for precision or space capacity as a function of the output length since this allows retaining a good practical approximation of computational power similar to that made when considering computers as Turing machines.
> Furthermore, there are language models that can adapt the computation time depending on factors such as the complexity of a task, meaning the unbounded computation time assumption of the upper bound may prove more relevant in the future.
> Upon acceptance, we will include both of these points in the discussion on potential future work.

---

### Official Review · Reviewer_EjFc · 2023-08-06

**Soundness:** 4

**Excitement:**

3: Ambivalent: It has merits (e.g., it reports state-of-the-art results, the idea is nice), but there are key weaknesses (e.g., it describes incremental work), and it can significantly benefit from another round of revision. However, I won't object to accepting it if my co-reviewers champion it.

**Missing References:**

* Rabin, Michael O. "Real time computation." Israel Journal of Mathematics 1, no. 4 (1963): 203-211.
* Gill III, John T. "Computational complexity of probabilistic Turing machines." In Proceedings of the sixth annual ACM symposium on Theory of computing, pp. 91-95. 1974.

**Paper Topic And Main Contributions:**

This paper explores the computational power of language models based on recurrent neural networks (RNNs). In particular, it extends the Turing completeness result to the probabilistic case, demonstrating that a recurrent language model with rational weights and unlimited computation time can simulate any probabilistic Turing machine.

Since real-world RNN language models process one symbol at a time in real-time, the above result is considered the maximum expressiveness of RNN-based language models. The paper further establishes a lower bound by demonstrating that these models can simulate deterministic real-time rational probabilistic Turing machines under the constraint of real-time computation.

In summary, the paper establishes that RNN-based language models with rational weights are as powerful as probabilistic Turing machines when computation time is unbounded. However, in practical real-time scenarios, their expressive power is limited to simulating deterministic real-time rational probabilistic Turing machines.

**Questions For The Authors:**

Question A. In light of the Turing completeness findings from Chung and Siegelmann (2021) regarding recurrent neural networks (RNNs) with finite precision and the exciting results presented by Merril et al. (2019-2022) on the capabilities and limitations of saturated RNNs, I am curious to understand the unique insights offered by your work. I ask this question with a genuine intention to ensure that I have not overlooked any novel perspectives your paper might bring, because as it stands, I am afraid that the current work does not seem to introduce new insights to the field.

**Reasons To Accept:**

* The paper focuses on an important problem in the study of language models, namely the computational expressivity of language models based on (Elman-type) recurrent neural networks (RNNs). It extends the Turing completeness result of RNNs to a probabilistic context, demonstrating that RNN-style language models with rational weights and arbitrary computation time can simulate a probabilistic Turing machine (PTMs). Moreover, the paper shows that these models, under real-time computation constraints, can simulate deterministic real-time PTMs.
* The paper is well-written and relatively easy to follow. The proofs presented in the paper are also sound and convincing.
* The first half of the paper provides a clear exposition of classical results in the study of Turing machines.

**Reasons To Reject:**

* The unique contributions of the paper are not well-defined and difficult to discern. The authors claim to demonstrate that RLMs with rational weights and arbitrary computation time can capture the same probability distributions over strings as probabilistic Turing machines. However, this result does not appear to be surprising or novel.
* On multiple occasions, the authors emphasize the fact that their analysis is on probability distributions rather than unweighted languages, but they fail to adequately motivate or explain the significance of this distinction. As a result, the entire analysis presented in the paper feels rather abstract and obscure.
* The paper extensively covers well-known concepts and results up to its final page. However, to enhance its potential for publication, it should address Open Question 5.1 and delve into the precise computational power of a rationally weighted recurrent language model. This addition would provide valuable insights and contribute significantly to the existing literature.
   * At present, the paper lacks original findings or conclusive answers to the questions it raises. To truly captivate readers, the authors should consider exploring the computational power of real-time recurrent or self-attention-based language models with finite precision and rational hidden states. Such an investigation would introduce a refreshing and exciting dimension to the study, elevating its significance and making it a valuable contribution to the field.

**Reproducibility:**

5: Could easily reproduce the results.

**Reviewer Confidence:**

4: Quite sure. I tried to check the important points carefully. It's unlikely, though conceivable, that I missed something that should affect my ratings.

**Typos Grammar Style And Presentation Improvements:**

* The purpose and significance of Figure 1 are unclear. It does not seem to contribute or enhance the overall understanding of the paper.
* You might want to move Figure 2 to the first page.
* L142: Is the word “novel” necessary in that sentence? You are not reinventing probabilistic Turing machines again.
* L157: Why did you opt for the symbol $\sigma$ to represent a sequence of text? Wouldn't $y$ (or $x$) be more suitable in this context, considering that some readers might associate $\sigma$ with the sigmoid function?

---

> ### Author Rebuttal · Authors · 2023-08-29
>
> Thank you very much for your detailed and thoughtful review. Thank you also for the additional references you suggested, we shall add them to the manuscript.
>
> Regarding your first point, thank you for pointing out that our contributions should be clarified in the introduction. To also answer your related point from the typos, grammar, and style section:
> We do not claim to be reinventing probabilistic Turing machines. However, the distinction between a deterministic Turing machine, a probabilistic Turing machine, and a deterministic probabilistic Turing machine, especially in the context of real-time computation, has not been made before, at least to our knowledge. We will include an additional discussion to the paper relating all these different types of Turing machines to the types of distributions they can express, as well as their computational expressivity.
>
> Regarding your second point, we think that we have provided motivation in the Introduction section, where we point out that “LMs inherently work with probability distributions and do not simply decide language membership” (lines 34-41). Hence, our focus on the probabilistic case does not only more adequately fit the language models used in practice, but also allows us to relate this line of work to “the measure-theoretic foundations of LMs (Welleck et al., 2020; Meister et al., 2023; Du et al., 2022)”, see lines 125-130.
> Furthermore, as one of the references you mentioned points out, probabilistic Turing machines define different complexity classes to standard Turing machines. While it is still an unsolved problem to show whether or not e.g. the complexity classes P and RP are distinct, showing that RNN LMs can simulate any PTM in real-time is still a distinct, and potentially stronger, result.
>
> Regarding your last point, and also to answer your Question A at the same time:
> Our unique contribution is twofold: 1) extend Siegelmann and Sontag’s result, showing that RNN language models can not just simulate TMs but PTMs in real-time, generally regarded as a different and potentially stronger (in terms of complexity classes) computational model which is also better suited to theoretically explain language models.
> 2) We provide a lower bound under more realistic conditions of real-time computation. In contrast to Merril’s work on saturated RNNs, which shows that RNNs with finite precision are limited to recognizing regular languages, we show that relaxing the precision requirement allows for a lower bound that is much more expressive than finite state automata, namely, real-time deterministic Turing machines.
>
> Thank you for your pointers regarding representations of strings and symbols in our script, as well as the figures. We will change the symbols following your suggestion and change Figure 1.

---

### Official Review · Reviewer_BtV4 · 2023-08-07

**Soundness:** 4

**Excitement:**

4: Strong: This paper deepens the understanding of some phenomenon or lowers the barriers to an existing research direction.

**Paper Topic And Main Contributions:**

Siegelmann and Sontag (1992) established the result that RNNs with rational weights and unbounded computation time are Turing complete. This work extends the result to the probabilistic case, showing that such an RNN LM can simulate any probabilistic Turning machine. Moreover, this work also defines more realistic real-time rational PTMs and draw the connection between RLMs with such constraints with their PTM counterparts.

The paper is well-written. Despite its abstract nature, it is easy to follow. The work brings the result of 1992 closer to the realistic case of  language modeling.

**Questions For The Authors:**

Along the line of imposing realistic assumptions in analyzing expressivity, do you have insights on how bounded-precision and/or fixed network dimensions (such as number of layers and number of cells) affect expressivity?

**Reasons To Accept:**

It is a well-written paper that brings the formal language result of Siegelmann and Sontag (1992) one step closer to the language modeling realities.

**Reasons To Reject:**

The paper does not quite fit the main focus of the conference which is the empirical aspect of natural language processing. From the theoretical perspective, the extension from the Turing-complete problem to the probabilistic case is incremental rather than fundamental.

**Reproducibility:**

3: Could reproduce the results with some difficulty. The settings of parameters are underspecified or subjectively determined; the training/evaluation data are not widely available.

**Reviewer Confidence:**

2: Willing to defend my evaluation, but it is fairly likely that I missed some details, didn't understand some central points, or can't be sure about the novelty of the work.

---

> ### Author Rebuttal · Authors · 2023-08-29
>
> Thank you very much for taking the time to understand our work and for your review.
>
> We appreciate that the topic of our paper is theoretical in nature. However, EMNLP has always included theoretical works alongside experimental ones. Furthermore, we believe that understanding the theoretical aspects of the models used by empirical research on a formal level will benefit researchers in improving the capabilities of these models beyond what they are capable of generating now.
>
> Regarding the step from Turing machines to probabilistic Turing machines being incremental, we would argue that it is, in fact, an important distinction to make, as modern language models do not operate as recognizers at all, simply accepting or rejecting strings. Instead, their ability to approximate probabilities of words and strings in context is integral to their functioning. Hence we contend that probabilistic Turing machines are a (much) more suitable and nuanced formal model, motivating our focus on proving their connection to recurrent language models.
>
> Thank you for your question regarding more realistic assumptions of precision and model parameters. As we noted in the introduction, finite precision means in practice the capacity of RNNs is limited to that of finite-state machines.
> In the Turing machine formulation, we chose to stick with infinite precision, in line with the classical paper by Siegelmann and Sontag. However, it is also possible to bound the computation space (precision and number of hidden neurons) in relation to the string length, e.g. imposing precision logarithmic in the length. While this was outside the scope of this work, it presents an interesting direction for future work since it might yield a tighter upper bound.

---

### Meta-Review · Area_Chair_HLj1 · 2023-09-14

**Recommendation:** 4

**Metareview:**

Pros:
- A strong theoretical result that RNN LMs can simulate probabilistic Turing machines.
- All reviewers agree the paper is exceptionally well-written, especially for a theory paper.

Cons:
- The result is incremental, extending an existing finding to a probabilistic rather than weighted model.

---

### Decision · Program_Chairs · 2023-10-07

**Decision:**

Accept-Main

**Comment:**

Pros:
- A strong theoretical result that RNN LMs can simulate probabilistic Turing machines.
- All reviewers agree the paper is exceptionally well-written, especially for a theory paper.

Cons:
- The result is incremental, extending an existing finding to a probabilistic rather than weighted model.